# Systematic lncRNA mapping to genome-wide co-essential modules uncovers cancer dependency on uncharacterized lncRNAs

**Ramkrishna Mitra, Clare M Adams, Christine M Eischen\***

Department of Cancer Biology, Sidney Kimmel Cancer Center, Thomas Jefferson University, Philadelphia, United States

**Abstract** Quantification of gene dependency across hundreds of cell lines using genome-scale CRISPR screens has revealed co-essential pathways/modules and critical functions of uncharacterized genes. In contrast to protein-coding genes, robust CRISPR-based loss-of-function screens are lacking for long noncoding RNAs (lncRNAs), which are key regulators of many cellular processes, leaving many essential lncRNAs unidentified and uninvestigated. Integrating copy number, epigenetic, and transcriptomic data of >800 cancer cell lines with CRISPR-derived co-essential pathways, our method recapitulates known essential lncRNAs and predicts proliferation/growth dependency of 289 poorly characterized lncRNAs. Analyzing lncRNA dependencies across 10 cancer types and their expression alteration by diverse growth inhibitors across cell types, we prioritize 30 high-confidence pan-cancer proliferation/growth-regulating lncRNAs. Further evaluating two previously uncharacterized top proliferation-suppressive lncRNAs (*PSLR-1*, *PSLR-2*) showed they are transcriptionally regulated by p53, induced by multiple cancer treatments, and significantly correlate to increased cancer patient survival. These lncRNAs modulate G2 cell cycle-regulating genes within the FOXM1 transcriptional network, inducing a G2 arrest and inhibiting proliferation and colony formation. Collectively, our results serve as a powerful resource for exploring lncRNA-mediated regulation of cellular fitness in cancer, circumventing current limitations in lncRNA research.

**\*For correspondence:**
christine.eischen@jefferson.edu

## Editor's evaluation

This manuscript makes a valuable contribution to the common understanding of the function of lncRNAs in cancer formation and progression. Besides developing and applying a robust analysis framework of large-scale pan-cancer omics datasets to discover the roles of 30 long non-coding RNAs (lncRNAs) in cancer proliferation and growth, the authors performed direct function-testing experiments to validate the predicted biological mechanisms of two lncRNAs. The analysis framework developed here can serve as a resource to study the functions of lncRNA in cancer, and the computational framework can also be further extended to study cancer-relevant transcriptional and post-transcriptional regulation.

## Introduction

Sequencing data estimate the existence of >1,00,000 long noncoding RNAs (lncRNAs) in the human genome (*Zhao et al., 2021*). lncRNAs are RNAs longer than 200 nucleotides that do not encode proteins, but have emerged as critical modulators of gene expression networks through diverse regulatory mechanisms, including regulation of RNA transcription, chromatin remodeling, and nuclear

architecture (*Statello et al., 2021*). Despite the expression of a large fraction of the lncRNAs being altered in diseases, including cancer, the functions of only a very small fraction have been verified, and even fewer are known for their roles in regulating fundamental cancer Hallmark processes, such as cell proliferation/growth (*Bao et al., 2019*; *Liu et al., 2017b*; *Ning et al., 2016*; *Schmitt and Chang, 2016*). Dysregulated cell proliferation/growth is at the heart of cancer, and, thus, identifying the genes and RNAs that regulate it is critical.

A key advance towards identifying essential proliferation/growth regulators was the development of RNA interference and genome-scale clustered regularly interspaced short palindromic repeats (CRISPR)–Cas9 screens (*Blomen et al., 2015*; *Hart et al., 2015*; *McFarland et al., 2018*; *Meyers et al., 2017*; *Tsherniak et al., 2017*). These approaches were used for >700 cancer cell lines and identified >1800 pan-cancer proliferation/growth and/or survival-regulating essential genes (*Blomen et al., 2015*; *Hart et al., 2015*; *Tsherniak et al., 2017*). Such loss-of-function screens for lncRNA have lagged significantly behind protein-coding genes and lack sufficient robustness. Only seven cell lines were screened for growth and/or survival modulating lncRNA loci (*Liu et al., 2017a*). In another study, CRISPR screening of keratinocytes determined nine lncRNAs that regulate its proliferation (*Cai et al., 2020*). Although lncRNAs are emerging as critical cell proliferation/growth regulators, a very limited number of screens that covered only a few cell lines has left many potentially essential lncRNAs unidentified and uninvestigated in cancer.

Increasing evidence suggests that lncRNAs are part of gene regulatory units/modules, where one or more lncRNAs regulate complex physiological processes, including those required in cancer, by modulating the expression of protein-coding genes (*Schmitt and Chang, 2016*; *Statello et al., 2021*). Over 5000 co-essential gene complexes/modules have been identified from genome-wide CRISPR screens of 485 cell lines of multiple different cancer types (*Wainberg et al., 2021*). The identified modules recapitulate diverse known pathways and protein complexes that have critical oncogenic functions during tumorigenesis. Since functionally related genes are highly enriched in co-essential modules (*Wainberg et al., 2021*), here we hypothesize that a statistically robust mapping of lncRNAs to genome-wide co-essential modules should enable unbiased predictions of proliferation/growth dependencies of uncharacterized lncRNAs and circumvent the limitations of lncRNAome-wide knockout/activation screens.

With the goal of mapping lncRNAs to co-essential modules systematically, we developed a large-scale data analysis framework. Our integrative computational framework consistently prioritizes known proliferation/growth inducers and suppressors as the top hits in both the evaluation and validation datasets. Additionally, we predict proliferation/growth dependencies of 289 previously uncharacterized lncRNAs. We experimentally validated transcriptional regulation and functions of top predicted proliferation/growth suppressor lncRNAs in lung adenocarcinoma (LUAD), whose increased expression correlates with favorable overall survival across cancer types. Our study significantly advances the ability to decode high-confidence critical lncRNAs upon which cancer cells are dependent for their growth and survival, accelerating further biological discovery of lncRNAs.

## Results

### Mapping lncRNAs to genome-wide co-essential modules reveals critical uncharacterized lncRNAs in cancer

We devised an approach based on a multivariate regression model, a classic statistical technique, to map the lncRNAome in genome-wide cancer cell proliferation/growth-regulating co-essential gene networks (*Figure 1*). This model estimated lncRNA-gene co-expression networks accounting for the noise from DNA methylation and copy number alterations, two factors that introduce false positives in gene regulatory networks (*Figure 1A*; 'Materials and methods'; *Jacobsen et al., 2013*; *Mitra et al., 2020*; *Mitra et al., 2017*). We applied the multivariate model to our evaluation data containing molecular profiles of 807 cancer cell lines in the cancer cell line encyclopedia (CCLE) (*Ghandi et al., 2019*) and 10 validation datasets consisting of 3017 patient samples of 10 TCGA cancer types (*Hoadley et al., 2018*; *Figure 1A*; *Figure 1—figure supplements 1 and 2*).

From the CCLE-based lncRNA-gene networks, we identified 1049 (6.6%) of the GENCODE (V27) (*Frankish et al., 2019*) annotated lncRNAs (n = 15777) that have significantly enriched (Benjamini–Hochberg [BH] adjusted p<$10^{-10}$; hypergeometric test) positive (549 lncRNAs; *Supplementary file*

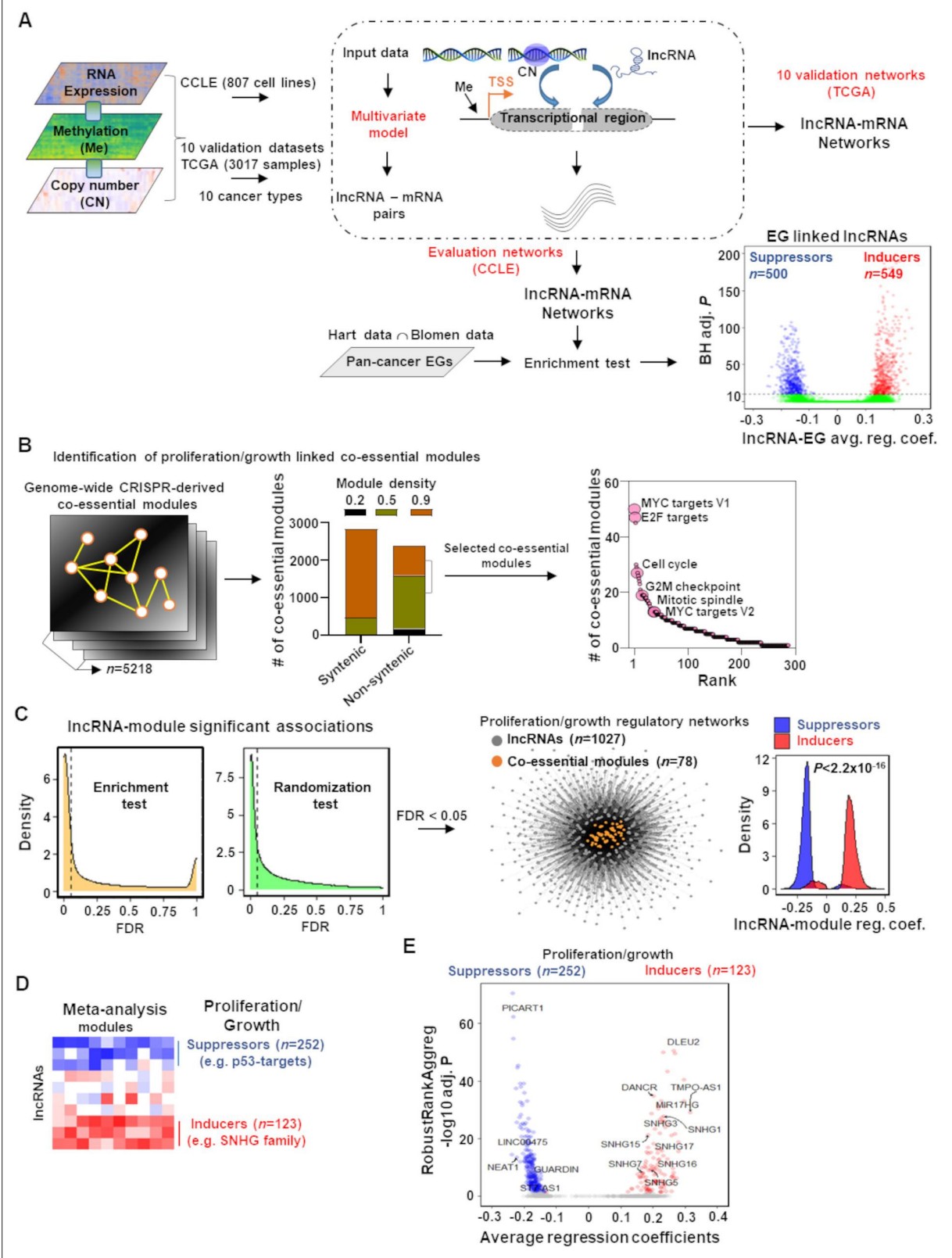

**Figure 1.** Mapping of long noncoding RNAs (lncRNAs) to genome-wide co-essential networks. (**A**) Identification of lncRNA-mediated mRNA expression changes after controlling for the impact of DNA methylation and gene copy number changes by modeling a multivariate regression function. The model was separately employed for pan-cancer cell lines (evaluation data) and patient samples (validation data) of 10 TCGA cancer types. The lncRNAs showing significantly enriched associations with the essential genes were identified (*n* denotes the number of lncRNAs; EG: essential genes; BH adj. *P*

*Figure 1 continued*

denotes Benjamini–Hochberg corrected adjusted hypergeometric p-value; avg. reg. coef. denotes regression coefficients; ∩: intersection). (**B**) Selection of high-confidence proliferation/growth-regulating modules (*n* denotes number of co-essential modules). (**C**) The lncRNAs and co-essential modules with significant association strengths (left of dashed vertical lines; FDR: false discovery rate), measured by indicated statistical tests, were selected to construct lncRNA-mediated proliferation/growth-regulatory networks. *n* denotes the number of indicated node types present in the network. Density plot (center) indicates distribution of lncRNA-module positive or negative associations. p-Value in the density plot was calculated from two-tailed Wilcoxon rank-sum test. Reg. coef. denotes regression coefficients. (**D**) A schematic heatmap showing indicated numbers of proliferation/growth-inducing (red) or suppressing (blue) lncRNAs, identified from the meta-analysis tool RobustRankAggreg. (**E**) Volcano plot of the lncRNAs that potentially induce (red dots) or suppress (blue dots) proliferation/growth-regulating co-essential modules. X-axis summarizes lncRNA-module association strength where positive and negative scores indicate the lncRNAs potentially induce or suppress the co-essential modules, respectively. RobustRankAggreg significance levels, denoted on Y-axis, indicate the lncRNAs with consistent positive (red dots) or negative (blue dots) associations across the co-essential modules or the remaining lncRNAs (gray dots).

The online version of this article includes the following source data and figure supplement(s) for figure 1:

**Source data 1.** Long noncoding RNAs (lncRNA)-mediated potential regulation of co-essential modules.

**Figure supplement 1.** Ten TCGA cancer types evaluated in this study.

**Figure supplement 2.** Distribution of samples having each of the four different types of omics data in each of the 10 TCGA cancer types.

**Figure supplement 3.** The number of co-essential modules associated with proliferation or other gene signatures.

**Figure supplement 4.** Percentage of long noncoding RNA (lncRNA)-associated co-essential modules showing enrichment with proliferation or other gene signatures.

*1a*) or negative (500 lncRNAs; *Supplementary file 1b*) associations with the pan-cancer cell survival/growth-regulating essential genes (EG; *Figure 1A*; 'Materials and methods'). The positive or negative associations with a stringent significance cutoff indicate high confidence that these lncRNAs induce or suppress survival/growth, respectively.

We systematically mapped these 1049 lncRNAs to predefined genome-wide co-essential pathways/modules (*Wainberg et al., 2021*). To retain high-quality co-essential modules, we rejected syntenic ones (i.e., all genes within the module on the same chromosome) and selected only those that have higher (≥0.5) module density scores (*Wainberg et al., 2021*; *Figure 1B*; 'Materials and methods'). As there is no report of the impact of these modules on cancer Hallmark processes, we reannotated their functions using Hallmark genesets (*Liberzon et al., 2015*) and Kyoto Encyclopedia of Genes and Genome (KEGG) (*Kanehisa and Goto, 2000*) annotated pathways ('Materials and methods'; *Supplementary file 1c and d*). As expected, the co-essential modules were preferentially (p=2.06 × $10^{-4}$; Wilcoxon rank-sum test) enriched with proliferation/growth-regulating genesets (MYC targets V1 and V2, E2F targets, G2/M checkpoint, cell cycle, and mitotic spindle) compared to other genesets/pathways (*Figure 1B*, *Figure 1—figure supplement 3*, *Supplementary file 1e*). Using two independent statistical tests (hypergeometric-based enrichment analysis and a thousand randomization test), we estimated whether an lncRNA regulates a module with higher confidence ('Materials and methods'; *Figure 1C*). Both tests showed that the essential genes-linked lncRNAs have a greater association with proliferation/growth-regulating genesets/pathways compared to the other genesets/pathways (*Figure 1—figure supplement 4*). Among the 1049 lncRNAs, 1027 potentially regulate 78 proliferation/growth-linked co-essential modules (*Figure 1C*, *Supplementary file 1f*). The modules in the network were positively or negatively impacted preferentially by the essential genes inducing or suppressing lncRNAs, respectively (*Figure 1C*, *Figure 1—source data 1*). Adding direction into the regulation provides power to distinguish proliferation/growth-suppressing lncRNAs from those that induce this cancer hallmark process.

The regulating lncRNAs were ranked for each co-essential module based on their average regression coefficients with the module members. These rankings summarize lncRNA-module association strength. With a meta-analysis using RobustRankAggreg (v1.1) method (*Kolde et al., 2012*), we determined the lncRNAs that consistently rank higher as positive or negative regulators across the proliferation/growth-linked co-essential modules with statistical significance (*Figure 1D*; BH adjusted RobustRankAggreg p<0.05; 'Materials and methods'). The meta-analysis identified 123 and 252 lncRNAs that may induce or suppress cancer cell proliferation/growth by inducing or suppressing co-essential modules, respectively (*Supplementary file 1g and h*). Using a comprehensive PubMed search with the keywords 'lncRNA' and 'cancer,' we determined that over 66% of the identified lncRNAs have not been studied in cancer. Another 10% of the lncRNAs are poorly characterized as they have

only been mentioned in a single study, including those with exclusively prediction results (*Supplementary file 1i*). Collectively, our results reveal many unknown significant associations between lncRNA and cancer cell proliferation/growth-regulatory networks.

To validate our predictions, we evaluated lncRNAs from the small nucleolar RNA host gene (SNHG) family, previously recognized as biomarkers for cancer development and aggressiveness (*Jin et al., 2019*; *Zimta et al., 2020*). Multiple members of the SNHG family of lncRNAs (e.g., *SNHG1*, *SNHG3*, *SNHG5*, *SNHG7*, *SNHG13/DANCR*, *SNHG15*, *SNHG16*, and *SNHG17*) have previously been shown to induce cancer cell proliferation/growth across different cancer types (*Figure 1E*; *Jin et al., 2019*; *Zimta et al., 2020*). Our meta-analysis predicted that these lncRNAs consistently positively associated with the co-essential modules, indicating elevated levels of these lncRNAs may induce cancer cell proliferation/growth. Furthermore, among our top predicted proliferation/growth inducers were the lncRNAs *DLEU2, TMPO-AS1*, and *MIR17HG*. *DLEU2* has been verified as a proliferation-inducing lncRNA in the vast majority of cancer types (13 out of 16) (*Ghafouri-Fard et al., 2021*), and *TMPO-AS1* has a role in tumorigenesis and progression and is overexpressed in 15 different cancer types (*Zheng et al., 2021*). The lncRNA *MIR17HG* is the host gene of the miR-17–92 cluster, which consists of a group of onco-microRNAs that induce pan-cancer cell proliferation/growth (*Mogilyansky and Rigoutsos, 2013*). In contrast, the tumor-suppressor p53 negatively impacts the essential genes and co-essential network modules (*Figure 1D*; *Wainberg et al., 2021*). The meta-analysis indicates known p53-regulated tumor-suppressor lncRNAs (e.g., *PICART1* [*Cao et al., 2017*], *LINC00475* [*Melo et al., 2016*], *NEAT1* [*Mello et al., 2017*], *LNCTAM34A/GUARDIN* [*Hu et al., 2018*], and *ST7-AS1* [*Sheng et al., 2021*]) were consistently negatively associated with the proliferation/growth-regulating co-essential modules (*Figure 1E*). Literature evidence suggests that elevated levels of these lncRNAs upon p53 activation suppress tumorigenesis (*Cao et al., 2017*; *Hu et al., 2018*; *Mello et al., 2017*; *Melo et al., 2016*; *Sheng et al., 2021*). Collectively, our approach recapitulates known proliferation/growth-regulating lncRNAs and predicts cancer cell proliferation/growth dependency of a large number (n = 289, >77%) of lncRNAs that are functionally poorly or not at all characterized.

## Pan-tumor tissues reproduce uncharacterized lncRNA-mediated growth regulation

To assess the impact of the cell line-derived proliferation/growth-linked lncRNAs in cancer patient samples, we evaluated lncRNA-mRNA associations independently in each of the 10 TCGA cancer types utilizing the same multivariate regression model ('Materials and methods'; *Figure 1A*). We separated the top 200 negatively and top 200 positively associated genes for each lncRNA and determined which of the cancer Hallmark gene signatures were significantly (BH adjusted p<0.05) enriched with genes that were positively or negatively associated with an lncRNA. We selected the top five gene signatures per lncRNA. Distribution of these lncRNAs across eight broad categories of Hallmark processes (*Liberzon et al., 2015*), constructed from the 50 Hallmark gene signatures, shows significant (p=1.96 × 10$^{-11}$; ANOVA) variability in the number of regulating lncRNAs. We determined that the proliferation/growth-linked lncRNAs, identified from the cell line-based evaluation data, preferentially regulate proliferation across TCGA cancer types (median = 142) compared to the other Hallmark processes (median range across the Hallmark categories = 15–99) (*Figure 2A*, *Figure 2—source data 1*). To investigate whether we biased the results by selecting the top 200 co-expressed protein-coding genes, we conducted Gene Set Enrichment Analysis (GSEA) (*Subramanian et al., 2005*), considering all the genes that have significant (BH adjusted p<0.001) association with each lncRNA to connect the lncRNA with cancer Hallmark processes. The top five significantly (BH adjusted p<0.05) enriched gene signatures were connected to each lncRNA. Again, significant (p=8.68 × 10$^{-14}$; ANOVA) variation in the number of regulating lncRNAs of the Hallmark processes across the TCGA cancer types was obtained (*Figure 2—figure supplement 1*). Importantly, proliferation was regulated by a higher number of lncRNAs (median = 187) compared to the other Hallmark categories (median range across Hallmark categories = 7–135) (*Figure 2—figure supplement 1*). Therefore, consistent biological interpretation observed from two different approaches that utilized different sets of genes indicates that the identified lncRNAs have a preferential association with proliferation/growth with high confidence.

We further investigated the direction of lncRNA-proliferation associations obtained from the top 200 lncRNA-gene pairs. We determined that the lncRNAs predicted as potential inducers and suppressors in the CCLE data showed significantly higher positive (5.17 × 10$^{-4}$; Wilcoxon rank-sum test) and

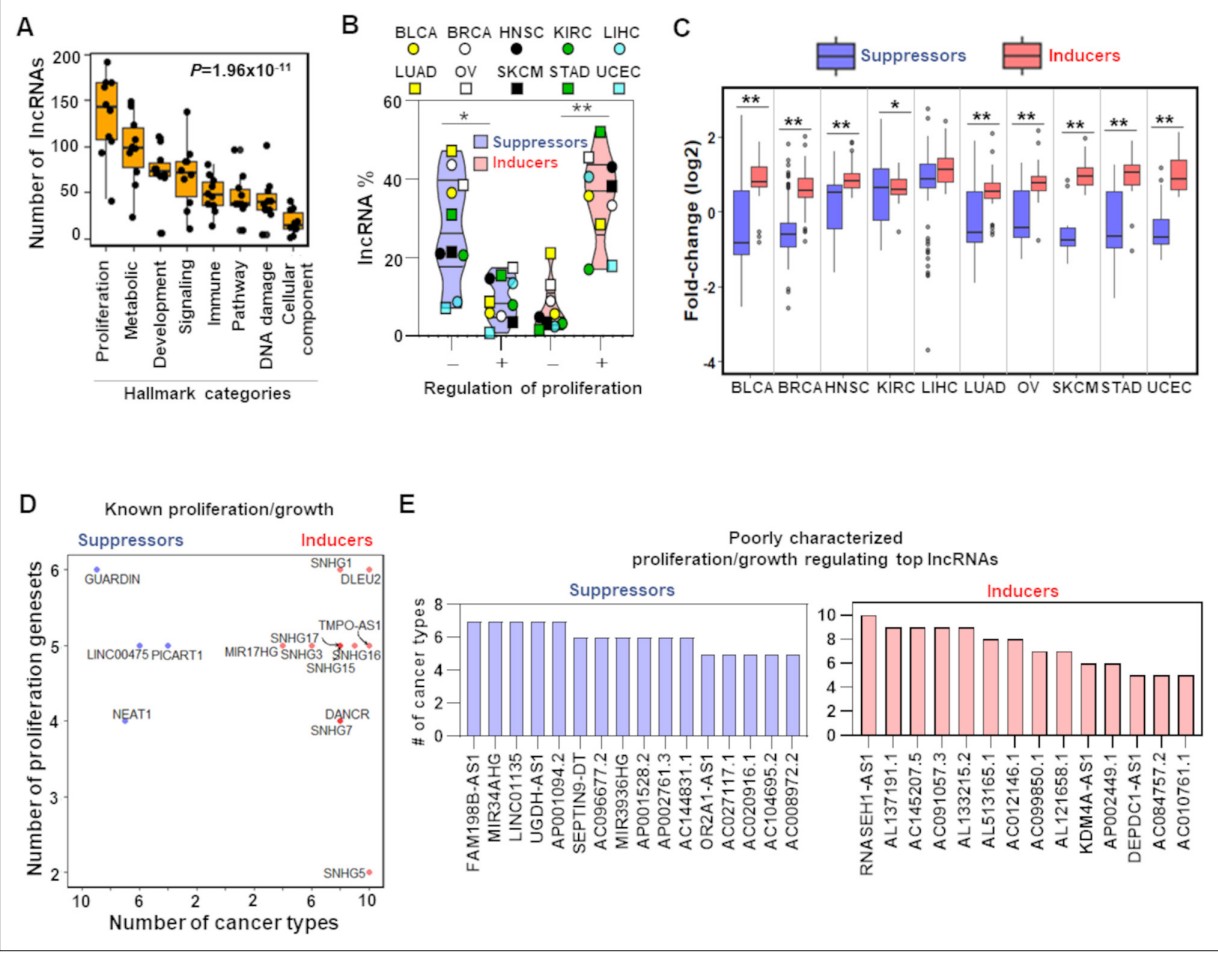

**Figure 2.** TCGA cancer patient samples reproduce growth-regulating long noncoding RNAs (lncRNAs). (**A**) Box plot shows indicated number of lncRNAs (Y-axis) that potentially regulate different Hallmark functions in cancer (X-axis) across 10 TCGA cancer types. p-Value calculated from ANOVA indicates a significant difference in the number of regulating lncRNAs across Hallmark functional categories. Dot represents individual cancer types. (**B**) Violin plots of the percentage of predicted proliferation-suppressing lncRNAs (blue) or proliferation-inducing lncRNAs (red) with negative (-) or positive (+) associations with proliferation signatures across TCGA cancer types; *p=3.19 × 10⁻³, **p=5.17 × 10⁻⁴; two-tailed Wilcoxon rank-sum test. (**C**) Levels of proliferation-inducing and -suppressing lncRNAs in patients with higher proliferation activity compared to patients with lower proliferation activity across 10 TCGA cancer types; *p=0.022, **p<1.68 × 10⁻⁶; two-tailed Wilcoxon rank-sum test. (**D**) Scatter plot shows known proliferation/growth-inducing and -suppressing lncRNAs that have positive or negative impact on the indicated number of Hallmark proliferation genesets (Y-axis) in the indicated number of TCGA cancer types (X-axis). (**E**) Poorly characterized lncRNAs that potentially regulate (induce or suppress) proliferation/growth in at least five Hallmark proliferation genesets in five or more TCGA cancer types are shown.

The online version of this article includes the following source data and figure supplement(s) for figure 2:

**Source data 1.** The number of long noncoding RNAs (lncRNAs) predicted to regulate different Hallmark categories in a specific cancer type.

**Source data 2.** The number of cell line-based predicted proliferation-regulating long noncoding RNAs (lncRNAs) that potentially alter proliferation signatures across TCGA cancer types.

**Source data 3.** Cell line-based predicted proliferation-regulating long noncoding RNAs (lncRNAs) that have significant (false discovery rate [FDR] < 0.05) differential expression in TCGA patients with higher proliferation activity compared to the patients with lower proliferation activity.

**Figure supplement 1.** More long noncoding RNAs (lncRNAs) potentially regulate proliferation across TCGA cancer types.

negative (3.19 × 10⁻³; Wilcoxon rank-sum test) associations, respectively, with proliferation genesets across the cancer types (*Figure 2B*, *Figure 2—source data 2*). Altogether, the results indicate that cancer patient samples resemble the cell line data with respect to the identified lncRNA-mediated proliferation/growth regulation.

We then evaluated lncRNA levels in TCGA patient samples with higher proliferation activity compared to the samples with lower proliferation activity ('Materials and methods'). Predicted proliferation/growth inducers and suppressors had a significantly distinct distribution of expression

alterations in 9 out of 10 cancer types (*Figure 2C*; p<0.05, BH adjusted Wilcoxon rank-sum test; *Supplementary file 2a*, *Figure 2—source data 3*). The median fold change of proliferation inducers indicates their coordinated overexpression in all 10 TCGA cancer types, and the suppressors showed coordinated downregulation in 7 cancer types (*Figure 2C*). Therefore, the observed alterations of the proliferation/growth-inducing and -suppressing lncRNAs suggest they are selected for during tumorigenesis to allow for cancer cell proliferation/growth.

We also determined that 91% of known proliferation-inducing lncRNAs (highlighted in *Figure 1E*) have potentially induced proliferation/growth in a higher number of TCGA cancer types by regulating a higher number of Hallmark proliferation genesets (*Figure 2D*). Similarly, four of five known p53-regulated tumor-suppressor lncRNAs show a negative impact on a higher number of proliferation genesets across TCGA cancer types (*Figures 1E and 2D*). We determined 30 uncharacterized or poorly characterized lncRNAs that potentially regulate at least five Hallmark proliferation-linked genesets across five or more TCGA cancer types (*Figure 2E*). Their reproducible association with proliferation in both CCLE and TCGA datasets indicates they have high-confidence proliferation regulation potential that has not been addressed previously.

## Identifying proliferation/growth-regulating lncRNAs altered frequently by growth inhibitors

By analyzing seven RNA-seq datasets, we identified the proliferation/growth-regulating lncRNAs altered at least twofold in cells treated by one of the proliferation-inhibiting DNA-damaging stimuli (doxorubicin, 5-fluorouracil, gamma radiation) compared to control (*Supplementary file 2b*). We determined proliferation/growth-suppressing and -inducing lncRNAs were preferentially elevated and reduced, respectively (*Figure 3A*). These DNA-damaging proliferation inhibitors cause the activation of the p53 tumor-suppressor transcription factor. To pinpoint specific lncRNAs that have potential roles in p53-mediated tumor-suppressive functions, we analyzed seven additional RNA-seq datasets where p53 wild-type (p53$^{WT}$) cells were treated with Nutlin-3, a compound that blocks the binding of p53 to Mdm2 (*Vassilev et al., 2004*), triggering p53 activation independent of DNA damage, or controls ('Materials and methods'; *Supplementary file 2b*). Compared with the control cells, Nutlin-treated cells showed a preferential up- and downregulation (at least twofold) of the proliferation/growth-suppressing and -inducing lncRNAs, respectively, similarly to what we observed from the cells treated by the DNA-damaging proliferation inhibitors (*Figure 3A*). Furthermore, we observed reproducible expression patterns in Nutlin-treated single-cell RNA-seq profiles with p53 wild-type cells, but not for cells with mutant p53 (*Figure 3B*). Our results suggest that our identified proliferation/growth-regulating lncRNAs have notable crosstalk with p53-mediated tumor-suppressive pathways.

We determined that 7–32% (average = 17.2%) of the identified lncRNAs have more than twofold differential expression in individual RNA-seq datasets (*Figure 3—figure supplement 1*). The majority (59%) of lncRNAs showed differential expression in only one or two datasets, which supports cell type-specific functions of these lncRNAs (*Liu et al., 2017b*; *Figure 3—figure supplement 2*, *Supplementary file 1j*). Although high tissue/cell-type specificity is a reported feature of lncRNA, some critical lncRNAs are likely functional in many cancer types (*de Goede et al., 2021*; *Gao et al., 2021*). We determined that less than 5% of the lncRNAs were altered in more than five datasets (*Figure 3—figure supplement 2*, *Supplementary file 1j*). Conducting RobustRankAggreg-based meta-analysis, we identified 13 consistently upregulated lncRNAs across the RNA-seq datasets, and of them, 8 were identified as the top proliferation/growth suppressors (*Figure 3C*, *Supplementary file 1k*). We also identified four consistently downregulated lncRNAs and three of them were identified as the top proliferation/growth inducers (*Supplementary file 1l*). Consistent reduction or increased expression of these proliferation/growth-inducing or -suppressing, respectively, lncRNAs indicates their expression changes suppress the proliferation/growth of cells treated with the proliferation inhibitors and p53 activators. For example, a known pan-cancer onco-lncRNA, *TMPO-AS1*, was one of the top proliferation/growth-inducing lncRNAs and had more than twofold downregulation in six RNA-seq datasets (*Zheng et al., 2021*). Conversely, among the eight consistently upregulated lncRNAs, three are known p53-regulated tumor suppressors (*GUARDIN*, *PICART1*, and *LINC00475*) (*Cao et al., 2017*; *Hu et al., 2018*; *Melo et al., 2016*; *Figure 3C*), whereas the remaining lncRNAs have not been previously investigated for p53 regulation.

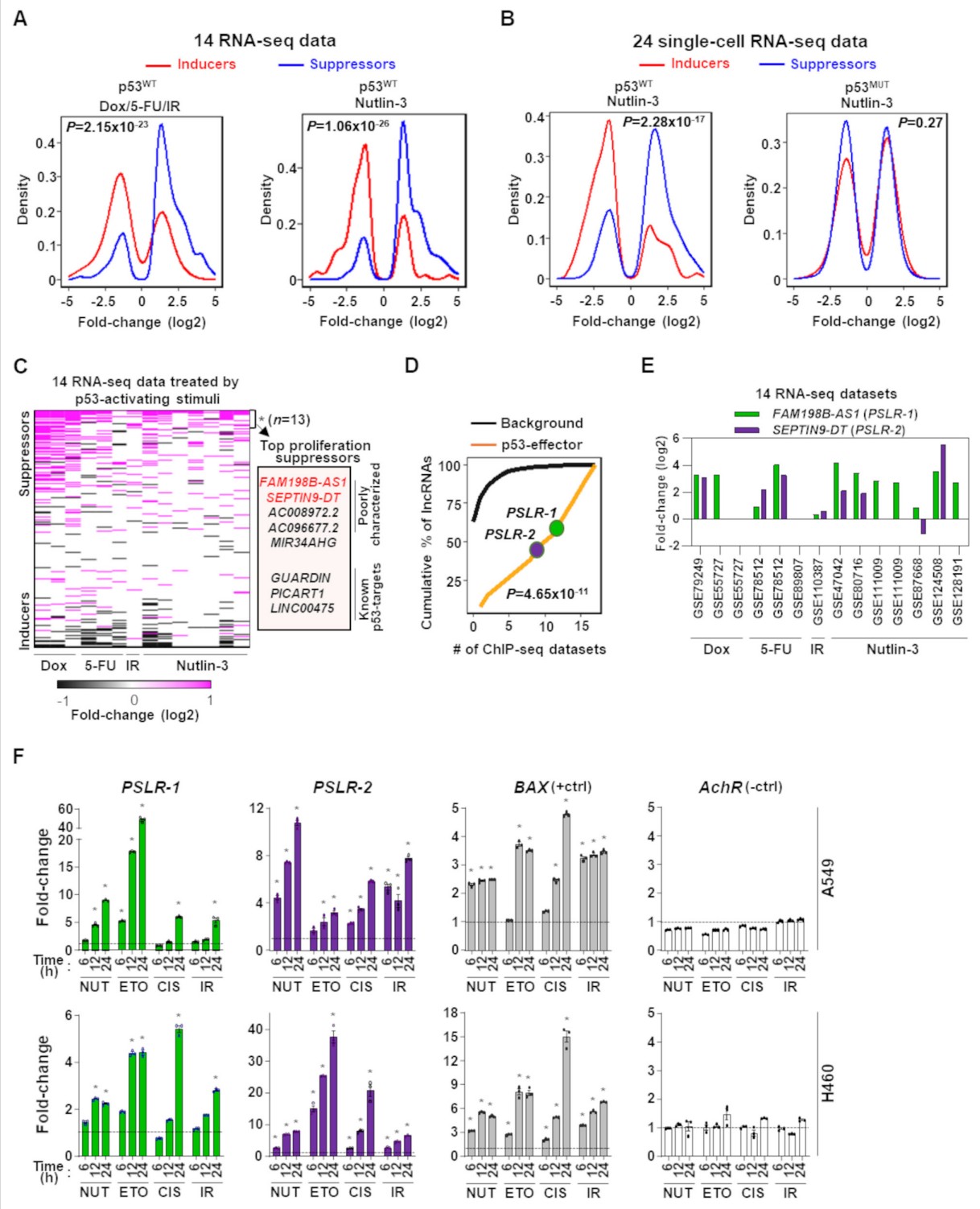

**Figure 3.** Modulation of growth-regulating long noncoding RNAs (lncRNAs) across diverse stresses. (**A**) Density plots showing the distribution of proliferation-inducing and -suppressing lncRNA levels in cells treated with doxorubicin (Dox), 5-fluorouracil (5-FU), or ionizing radiation (IR) in seven bulk RNA-seq data and seven bulk RNA-seq data of cells treated with Nutlin-3. (**B**) Density plots as in (**A**) from 24 single-cell RNA-seq data from cells treated with Nutlin-3 (7 p53$^{WT}$ and 17 p53$^{MUT}$ cell lines). (**C**) Heatmap showing fold change of predicted proliferation-inducing and -suppressing lncRNAs across 14 RNA-seq datasets, indicated in (**A**). *n* is the number of consistently upregulated lncRNAs across the RNA-seq datasets with statistical significance (Benjamini–Hochberg [BH] adjusted *p<0.05) calculated by the meta-analysis tool RobustRankAggreg; a subset of the top proliferation suppressors are highlighted in the box. (**D**) The number of ChIP-seq datasets showing p53-binding sites near the transcription start site of the 13 meta-analyses-derived

*Figure 3 continued on next page*

Figure 3 continued

significantly upregulated lncRNAs, indicated in (**C**), or background lncRNAs. For (**A**, **B**, **D**), p-values were calculated from two-tailed Wilcoxon rank-sum tests. (**E**) The expression fold change of two poorly characterized top predicted proliferation-suppressing lncRNAs in cells treated with p53-activating agents (see **A**) compared to control cells. The X-axis indicates the accession number of the 14 RNA-seq datasets present in the Gene Expression Omnibus database. (**F**) Two lung adenocarcinoma (LUAD) cell lines treated with 10 μM of Nutlin (NUT), etoposide (ETO), or cisplatin (CIS) or received 5 Gy of gamma-radiation (IR) were harvested at intervals. qRT-PCR (triplicate) for specific lncRNA was performed. *BAX* and *AchR* mRNA are positive and negative controls, respectively. RNA levels were normalized to *β-ACTIN,* and values are presented as $2^{-\Delta\Delta Ct}$, with vehicle control-treated cells set at 1 (black line); data are mean ± SEM. For A549, *PSLR-1* \*p<$6.86 \times 10^{-4}$, *PSLR-2* \*p<$1.17 \times 10^{-2}$, *BAX* \*p<$4.93 \times 10^{-6}$; for H460, *PSLR-1* \*p<$6.95 \times 10^{-6}$, *PSLR-2* \*p<$5.69 \times 10^{-3}$, and *BAX* \*p<$3.99 \times 10^{-4}$; two-tailed *t*-tests.

The online version of this article includes the following source data and figure supplement(s) for figure 3:

**Source data 1.** qRT-PCR for *Figure 3F*.

**Figure supplement 1.** Percentage of predicted proliferation/growth-regulating long noncoding RNAs (lncRNAs) showing at least twofold expression change in the cells treated with a growth inhibitor compared to the control cells.

**Figure supplement 2.** Distribution of differentially expressed long noncoding RNAs (lncRNAs) across RNA-seq datasets.

**Figure supplement 3.** p53 ChIP-seq datasets showing long noncoding RNAs (lncRNAs) bound by p53.

## p53 directly transcriptionally regulates proliferation/growth-suppressive lncRNAs

With the analysis of 17 p53 ChIP-seq datasets (*Supplementary file 2c*), we investigated whether the 13 lncRNAs that were consistently upregulated following treatment with proliferation inhibitors have the potential to be directly transcriptionally regulated by p53. We determined that a significantly (p=$1.65 \times 10^{-9}$; Wilcoxon rank-sum test) higher number of ChIP-seq datasets had p53-binding sites within 10 kilobases from the transcription start site of these lncRNAs compared with the remaining lncRNAs (background data) annotated in GENCODE version 27 (*Frankish et al., 2019*; *Figure 3D*, *Figure 3—figure supplement 3*). The two poorly characterized top proliferation suppressing lncRNAs (PSLRs) *FAM198B-AS1* (*PSLR-1*) and *SEPTIN9-DT* (*PSLR-2*) identified in our analysis had consistently elevated expression upon p53 activation across diverse cell types and treatments (more than twofold in at least six RNA-seq datasets; BH adjusted RobustRankAggreg p<$5.48 \times 10^{-3}$) (*Figures 2E, 3C and E*). These two lncRNAs also had p53-binding sites in at least nine ChIP-seq datasets (*Figure 3D*, *Figure 3—figure supplement 3*). To experimentally validate the meta-analysis results showing regulation of *PSLR-1* and *-2* expression by p53, we treated two p53^WT LUAD cell lines (A549 and H460) with p53-activating stimuli (Nutlin, etoposide, cisplatin, and gamma radiation) and evaluated their expression. Following exposure to any of the four p53-activating stimuli, levels of these two lncRNAs were significantly increased compared to vehicle control-treated cells (*Figure 3F*, *Figure 3—source data 1*). A known p53 target gene, *BAX*, and a gene that is not a p53 target, acetyl choline receptor (*AchR*), were positive and negative controls, respectively. These results support our bioinformatic analyses, showing activation of p53 results in increased expression of these lncRNAs. Notably, none of the 14 RNA-seq datasets used in the meta-analysis included lung cancer lines (*Supplementary file 2b*); therefore, our results from the high-throughput data analysis were validated in a completely independent cell type. These data indicate that previously uncharacterized lncRNAs are regulated by p53.

## *PSLR-1* and *-2* modulate CHR-enriched cell cycle-regulating pathways

To gain a greater understanding of *PSLR-1* and *-2*, we evaluated their expression levels in our own LUAD patient samples. Both lncRNAs were significantly (p<$1.23 \times 10^{-7}$; *t*-test) reduced in LUAD patient samples compared to normal lung tissue, indicating suppression of their functions during tumorigenesis (*Figure 4A*, *Figure 4—source data 1*). To experimentally verify the proliferation/growth-suppressive roles of the PSLRs in cancer cells and evaluate consequences independent of any drug effects that induce p53, we engineered A549 LUAD cells to express *PSLR-1*, *PSLR-2*, or vehicle control. From the lncRNA perturbation data, we determined that elevated expression of either of the two lncRNAs results in significantly (p<$7.9 \times 10^{-12}$; Wilcoxon rank-sum test) higher expression of those genes that had significant (BH adjusted regression p<0.001) positive associations with the lncRNAs in the TCGA LUAD patient cohort compared with the genes that had significant negative associations (*Figure 4B*). Further, we extracted curated tumor-suppressor genes and oncogenes from the literature (*Liu et al., 2017b*; *Zhao et al., 2016*) and the cancer gene census of the COSMIC database and selected a

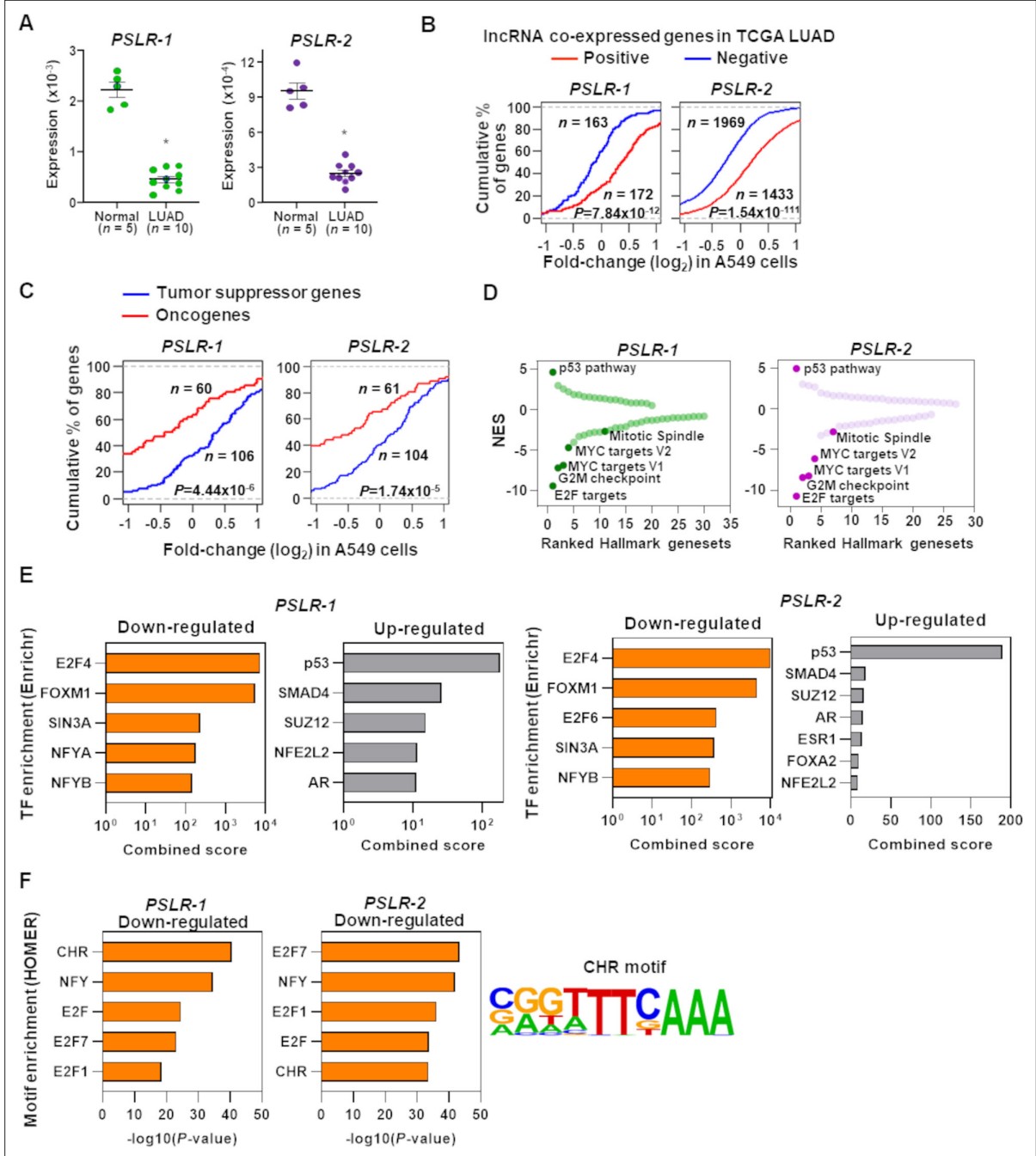

**Figure 4.** Functional characterization of *PSLRs* in lung adenocarcinoma (LUAD). (**A**) Expression of *PSLR-1* and *-2* was measured by qRT-PCR (triplicate) in normal human lung tissue (n = 5) and LUAD patient samples (n = 10). RNA levels were normalized to *β-ACTIN* and values are presented as $2^{-\Delta Ct}$; data are mean ± SEM. *PSLR-1*, *p=7.75 × $10^{-9}$; *PSLR-2*, *p=2.77 × $10^{-8}$; two-tailed *t*-tests. (**B–D**) RNA-sequencing of A549 LUAD cell line after expression of each of the two long noncoding RNAs (lncRNAs) or vector control (quadruplicate). (**B**) Genes that have significant (adjusted regression p<$10^{-3}$) positive or negative associations with the indicated lncRNA in TCGA LUAD data were selected. Empirical cumulative distribution of expression fold changes of these genes in A549 cells expressing the indicated lncRNA compared with vector control. (**C**) Empirical cumulative distribution of expression changes of curated tumor-suppressor genes and oncogenes in A549 cells expressing the indicated lncRNA compared with vector control. For (**B**, **C**), x-axis indicates the expression fold changes of genes, y-axis estimates the percentage of genes at a log2 fold-change value indicated on the x-axis, and p-values were from two-tailed Wilcoxon rank-sum tests. (**D**) Gene Set Enrichment Analysis (GSEA) demonstrated top up- and downregulated Hallmark genesets after expression of the indicated lncRNA in A549 cells compared with vector control. Positive and negative normalized enrichment scores (NES) indicate up- and downregulation, respectively. Significantly enriched (false discovery rate [FDR] < 0.05) proliferation-associated pathways/gene signatures indicated. (**E**) Enrichr-calculated combined score indicates transcription factor (TF) enrichment of genes altered upon expression of the indicated lncRNA in A549

*Figure 4 continued on next page*

*Figure 4 continued*

LUAD cells. (**F**) HOMER-calculated promoter motif enrichment of downregulated genes upon indicated lncRNA expression in LUAD cells. *CHR* motif elements are shown (right).

The online version of this article includes the following source data for figure 4:

**Source data 1.** qRT-PCR for *Figure 4A*.

subset of them as high confidence by determining whether they were significantly (≥2 fold change with BH adjusted p<0.05) down- or upregulated, respectively, in TCGA LUAD compared with normal lung samples. With elevated expression of each lncRNA in A549 cells, the tumor-suppressor genes and the oncogenes were preferentially elevated and suppressed, respectively (*Figure 4C*). Moreover, the distribution of their expression changes was significantly distinct (p<5.6 × 10$^{-6}$; Wilcoxon rank-sum test), indicating these lncRNAs act as negative regulators of lung tumorigenesis.

Following RNA-seq of our samples, we also evaluated transcriptome-wide changes by performing GSEA (*Subramanian et al., 2005*) to gain insight into which Hallmark genesets were significantly (FDR < 0.05) modulated with the increased expression of the two lncRNAs. The p53 pathway was consistently ranked as the top among the significantly (FDR < 0.05) upregulated Hallmark genesets in A549 cells with elevated levels of the two lncRNAs (*Figure 4D*). In contrast, the proliferation-linked Hallmark genesets (E2F targets, G2M checkpoint, mitotic spindle, and MYC targets V1 and V2) were also consistently ranked among the top significantly (FDR < 0.05) downregulated pathways/genesets, indicating they were negatively impacted with the overexpression of the two lncRNAs (*Figure 4D*). Collectively, our results indicate that previously unrecognized *PSLR-1* and *-2* likely suppress proliferation by negatively regulating proliferation-linked oncogenic genes and/or processes and positively regulating tumor-suppressive genes and/or processes.

Utilizing Enrichr (*Kuleshov et al., 2016*), a tool that integrates genome-wide ChIP experiments from ENCODE (*ENCODE Project Consortium, 2012*) and ChEA (*Lachmann et al., 2010*) projects, we investigated whether elevated PSLRs impact the expression of distal genes randomly or alter the transcriptional regulation of specific transcription factors. Increased *PSLR-1* or *-2* expression showed E2F4 and FOXM1 were the most enriched transcriptional regulators for the downregulated genes, and p53 was the top transcriptional regulator for the upregulated genes (*Figure 4E*). Subsequently, we identified regulatory motifs enriched in the promoters of the downregulated genes using the tool HOMER (*Heinz et al., 2010*). We determined that cell cycle genes homology region (*CHR*), a DNA element present in promoters of many cell cycle genes bound by E2F4 and/or FOXM1-containing protein complexes (*Chen et al., 2013*), was consistently among the top enriched motifs for both *PSLR-1* and *-2* (*Figure 4F*). Recent reports indicate that G2/M cell cycle genes with *CHR* sites are activated by a FOXM1-containing protein complex (*Chen et al., 2013*; *Sadasivam et al., 2012*). Our results from Hallmark GSEA, transcription factor, and motif enrichment analyses collectively indicate that elevated PSLRs likely modulate *CHR*-containing cell cycle genes within the E2F4 and/or FOXM1 transcriptional network, which may negatively impact G2/M cell cycle progression.

## *PSLR-1* and *-2* exert G2 cell-cycle arrest and suppress proliferation/growth in LUAD cells

To further verify whether *PSLR-1* and *PSLR-2* have growth-suppressive effects in cancer cells, we expressed them in two LUAD cell lines (A549 and H460) and evaluated the biological consequences. Expression of each lncRNA resulted in significantly decreased LUAD cell growth compared to cells with empty vector (*Figure 5A*, *Figure 5—source data 1*). Serving as an lncRNA control, we overexpressed *MALAT1*, a pro-proliferative lncRNA (*Tripathi et al., 2013*), which increased LUAD cell growth (*Figure 5A*, *Figure 5—source data 1*). Additionally, there was a decrease in the number of colonies formed from cells expressing *PSLR-1* or *PSLR-2* compared to cells with an empty vector or *MALAT1* (*Figure 5B*, *Figure 5—source data 2*). Because there was a significant decrease in live cell numbers with expression of these lncRNAs (*Figure 5C*, *Figure 5—source data 3*) without changes in cell viability (*Figure 5—figure supplement 1A*), we evaluated cell cycle. Expression of *PSLR-1* or *PSLR-2* resulted in increased numbers of cells in the G2/M phase of the cell cycle (*Figure 5D*, *Figure 5—source data 4*). To discriminate between cells in G2 and M phases, we evaluated phospho-histone H3, a marker of mitosis (*Hans and Dimitrov, 2001*). There was an

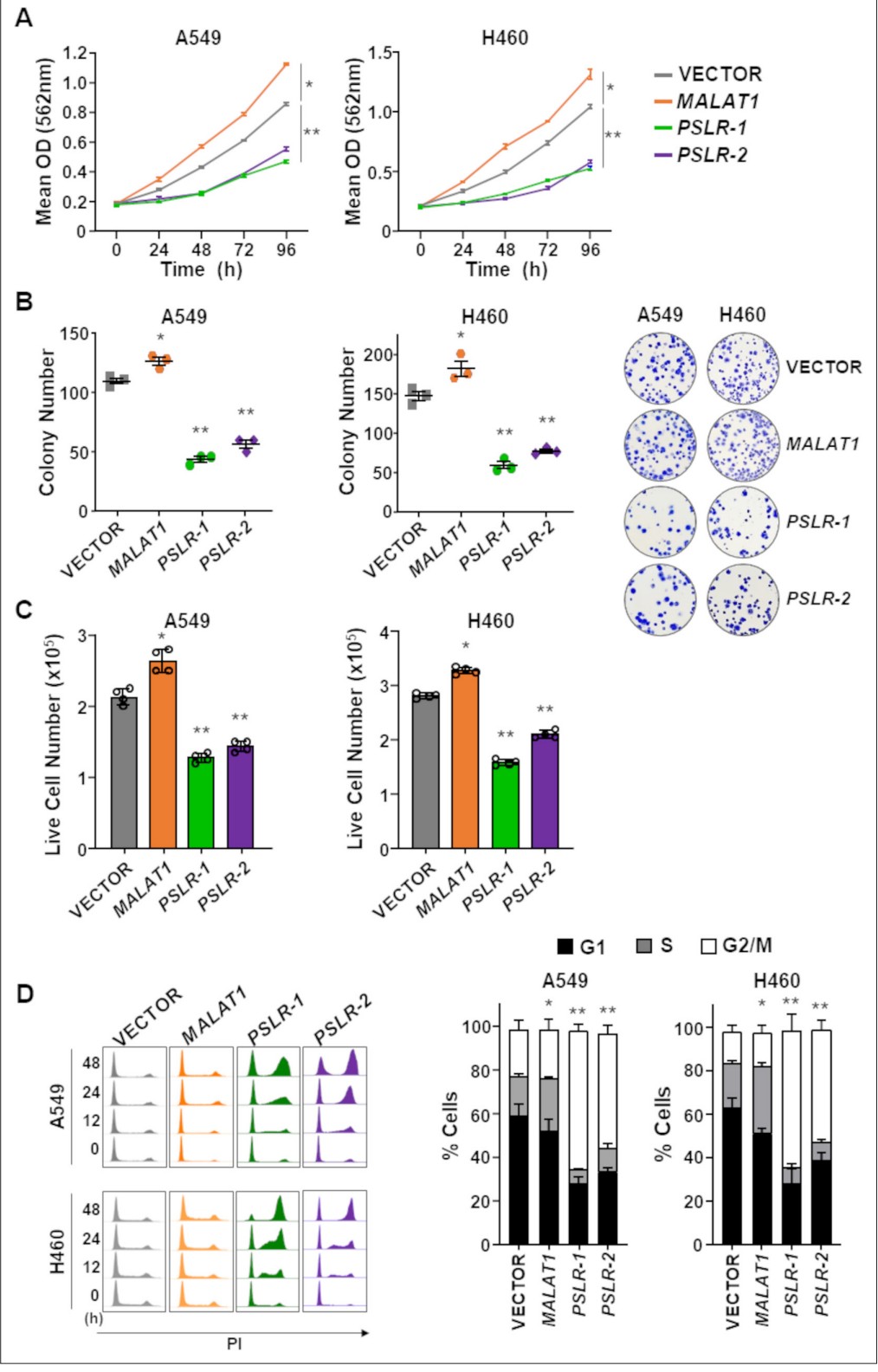

**Figure 5.** p53-regulated long noncoding RNAs (lncRNAs) exert tumor-suppressive activity in lung adenocarcinoma (LUAD). The lncRNAs *PSLR-1*, *PSLR-2*, or *MALAT1* or vector control were expressed (lentivirus) in LUAD cell lines. (**A**) MTT assays (quadruplicate) performed at 24 hr intervals. Each assay performed 2–4 independent times for both cell lines and one representative experiment is shown ± SD; A549 *p=1.11 × 10⁻⁴ and **p<3.03 × 10⁻⁴, H460 *p=2.04 × 10⁻⁵ and **p<2.28 × 10⁻⁴; two-tailed *t*-tests. (**B**) Colonies from colony formation assays were counted

*Figure 5 continued on next page*

*Figure 5 continued*

14 days after lncRNA expression. Assays (triplicate) performed 2–3 independent times and a representative experiment ± SEM and images (no magnification) shown; A549 *p=1.49 × 10⁻² and **p<2.09 × 10⁻⁴, H460 *p=3.42 × 10⁻² and **p<1.16 × 10⁻³; two-tailed *t*-tests. (**C**) Live cell number determined by Trypan Blue dye exclusion 48 hr after placing cells in 6-well plates. Mean of four independent experiments for both cell lines ± SD is graphed; A549 *p=5.77 × 10⁻⁴ and **p<2.68 × 10⁻⁵, H460 *p=2.90 × 10⁻⁵ and **p<3.63 × 10⁻⁶; two-tailed *t*-tests. (**D**) Cell cycle evaluated at intervals following propidium iodide staining and flow cytometry; representative histograms from one experiment shown (left). Graphs of the cell cycle phases represent the mean of four independent experiments for both cell lines at 48 hr ± SEM; A549 *p=8.82 × 10⁻³ and **p<1.73 × 10⁻², H460 *p=4.00 × 10⁻³ and **p<1.35 × 10⁻³; two-tailed *t*-tests.

The online version of this article includes the following source data and figure supplement(s) for figure 5:

**Source data 1.** MTT for *Figure 5A*.

**Source data 2.** Colony number for *Figure 5B*.

**Source data 3.** Live cell number for *Figure 5C*.

**Source data 4.** Cell cycle for *Figure 5D*.

**Figure supplement 1.** Cell viability and numbers of cells in mitosis are not altered by ectopic expression of the PSLRs.

---

analogous percentage of cells with phospho-histone H3 in cells expressing any of the lncRNAs or vector control (*Figure 5—figure supplement 1B*), indicating the cells were arresting in G2 and not M. Together, these data demonstrate a tumor-suppressive function of *PSLR-1* and *PSLR-2*, whereby an increase in their expression results in decreased cancer cell proliferation due to a G2 cell cycle arrest.

## *PSLR-1* and *-2* coordinately modulate the G2 cell cycle arrest network

To gain insight into the PSLR-induced G2 cell cycle arrest (*Figure 5D*), we determined from our RNA-seq data which genes in the G2 regulatory subnetwork that were indicated in the KEGG cell cycle pathway had significant alterations following increased expression of the PSLRs (*Figure 6A*). There was reduced expression of nine genes (blue squares; median fold change 7.67; median FDR = 2.27 × 10⁻⁹¹) and increased expression of three genes (red squares; median fold change 7.32; median FDR = 4.66 × 10⁻⁵²) with elevated PSLR levels, which would result in cells arresting in the G2-phase of the cell cycle (*Figure 6A*). To validate the changes in expression of these G2-related cell cycle genes, we performed qRT-PCR for seven of the genes in A549 and H460 cells expressing *PSLR-1*, *PSLR-2*, or vector control (*Figure 6B*, *Figure 6—source data 1*). Following forced expression of *PSLR-1* and *PSLR-2* mRNA levels of cell cycle regulators *CCNA2*, *CCNB1*, *CDC25C*, and *CDK1* were downregulated, whereas *CDKN1A*, *GADD45A*, and *SFN* (14-3-3-σ) were elevated in both lines (*Figure 6B*, *Figure 6—source data 1*). Furthermore, the expression changes observed at the RNA level for these genes were also reflected at the protein level following expression of the PSLRs (*Figure 6C*, *Figure 6—source data 2*) and are consistent with their function in G2. Altogether, our data indicate that the PSLRs coordinately alter the expression of the G2 cell cycle arrest network.

Because our multiple lines of evidence indicate that the PSLRs have proliferation/growth-suppressive functions in multiple cancer types through regulation of G2, we investigated alterations of the seven G2 cell cycle-regulating genes (*Figure 6B and C*) in the TCGA patient samples of the 10 cancer types (*Figure 1—figure supplement 1*). For individual cancer types, we divided the samples into two groups based on the highest or lowest quartile expression of *PSLR-1* or *PSLR-2*. We measured the expression changes of the G2 cell cycle-regulating genes in the patients with the highest quartile expression compared to the patients with the lowest quartile expression of the individual PSLRs. There was a significantly (p=5.09 × 10⁻⁴ for *PSLR-1* and P=2.72 × 10⁻⁴ for *PSLR-2*; Wilcoxon rank-sum test) distinct expression fold-change distribution of the genes that inhibit cell cycle arrest compared with the genes that induce the same process for both (*Figure 6D*). Altogether, our results provide a striking example of the power of our approach to accurately predict the functions of uncharacterized lncRNAs.

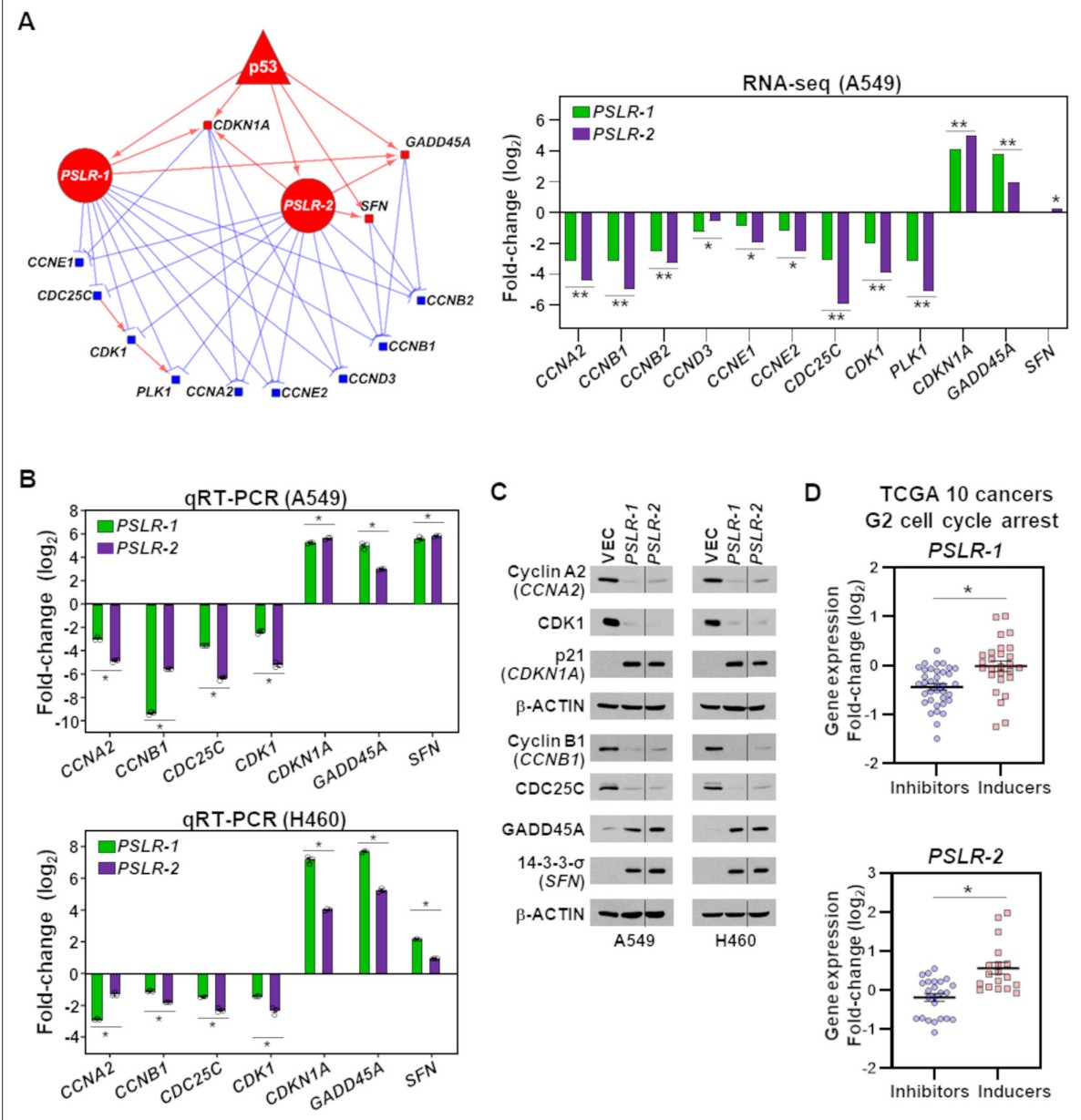

**Figure 6.** Elevated PSLR expression alters the G2 regulatory network across diverse cancer types. (**A**) A network was constructed from the genes that regulate cell cycle in G2 phase (left) and were significantly altered in PSLR-expressing A549 cells compared to control cells from our RNA-seq data (right). Circle, square, and triangle nodes represent the PSLR, genes, and p53, respectively. Red and blue nodes indicate up- and downregulation, respectively. The edges with direction indicate regulator-mediated target activation (red line with arrowhead) or suppression (blue line with T-shape head). The outgoing edges from the PSLRs indicate expression association. All other edges indicate biologically validated interactions. Benjamini–Hochberg (BH) adjusted *$p < 6.34 \times 10^{-3}$ and **$p < 3.23 \times 10^{-43}$ were calculated from EdgeR. (**B, C**) 48 hr after lentiviral expression of the *PSLRs* in lung adenocarcinoma (LUAD) cell lines, qRT-PCR (triplicate) for the cell cycle genes indicated was performed (**B**) and the proteins assessed by Western blotting (**C**). RNA levels were normalized to *β-ACTIN* and values plotted as log2 fold change relative to vector (VEC) control. Data are mean values ± SEM; A549 *$p < 2.55 \times 10^{-3}$ and H460 *$p < 5.31 \times 10^{-4}$; two-tailed *t*-tests. Vertical line in the Western blots denotes a lane was removed in between *PSLR-1* and *PSLR-2* of this exposure. (**D**) Expression fold-change distribution of the four genes that inhibit (blue dots) or the three genes that induce (red squares) G2 cell cycle arrest from (**B**) and (**C**). The expression fold changes were measured in samples with increased (highest quartile) PSLR expression compared to the samples with reduced (lowest quartile) PSLR expression. Accumulated results from 10 TCGA cancer types are shown; *$p < 2.73 \times 10^{-4}$ from two-tailed Wilcoxon rank-sum test denotes significantly distinct distribution.

The online version of this article includes the following source data for figure 6:

**Source data 1.** qRT-PCR for *Figure 6B*.

**Source data 2.** Western blots for *Figure 6C*.

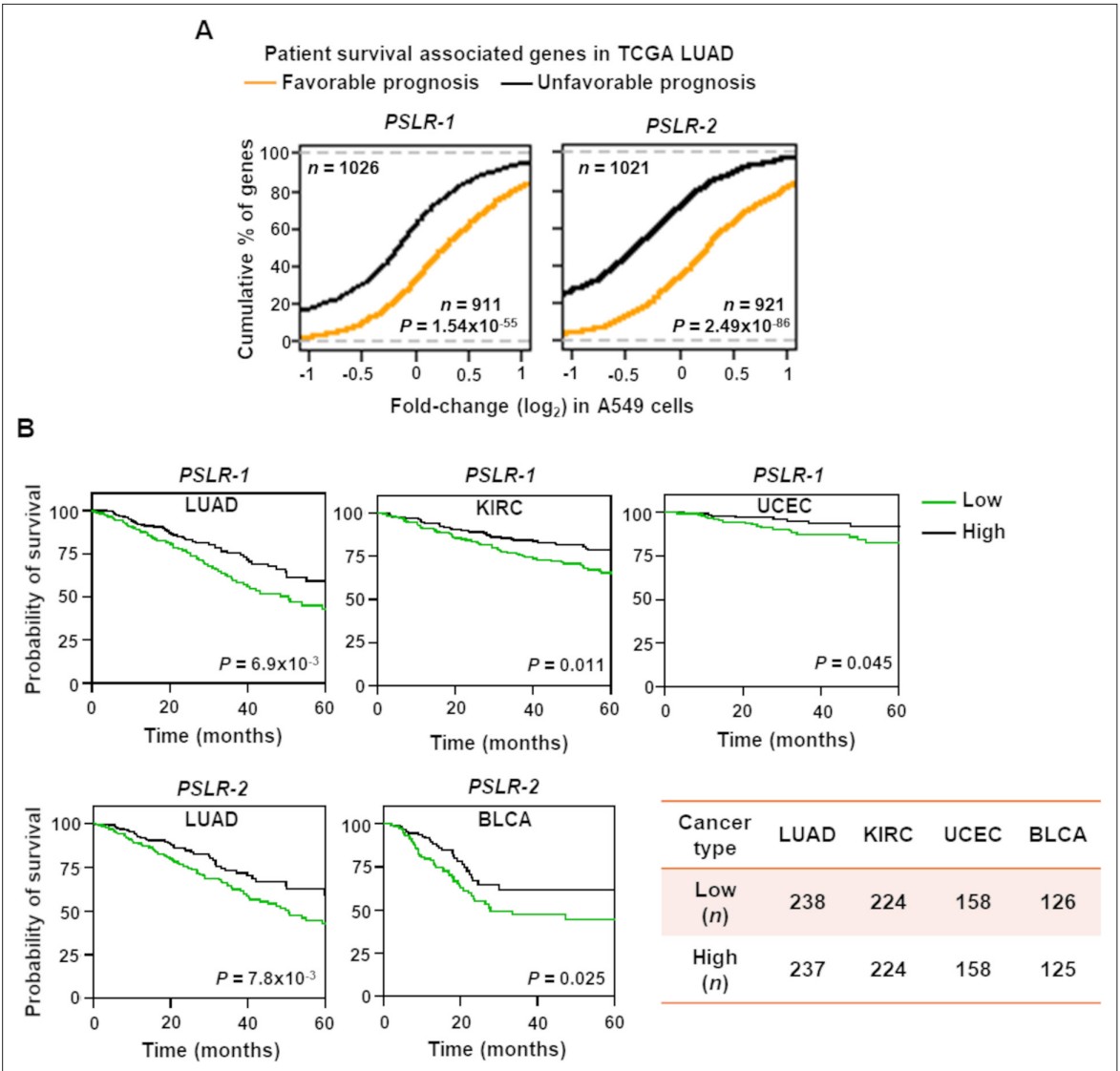

**Figure 7.** Elevated PSLR expression correlates with increased patient overall survival across diverse cancer types. (**A**) Distribution of expression changes of genes that have significant favorable or unfavorable prognostic associations with TCGA lung adenocarcinoma (LUAD) patient cohort; p-values from two-tailed Wilcoxon rank-sum tests. (**B**) Kaplan–Meier plots of overall survival associations between the indicated PSLR and the patients of the indicated cancer types. The low and high groups of patient samples were determined based on median expression of the indicated PSLR. To measure significantly different survival outcomes, log-rank tests were performed; *n* represents number of samples (**A, B**).

## Elevated *PSLR-1* and *-2* impact patient prognosis across cancer types

Given that the two PSLRs had proliferation/growth-suppressive functions, we evaluated whether their expression correlates with patient overall survival. First, utilizing our RNA-seq data where the individual PSLRs were ectopically expressed in A549 LUAD cells, we evaluated the distribution of the favorable and unfavorable prognostic genes of the TCGA LUAD patient cohort. The genes whose overexpression correlated with favorable patient outcome had preferentially higher expression upon elevated levels of each lncRNA in A549 cells. In contrast, elevated PSLR expression preferentially reduced those genes whose overexpression is linked with unfavorable prognosis (*Figure 7A*). Significant (p=1.54 × 10$^{-55}$ for *PSLR-1* and p=2.49 × 10$^{-86}$ for *PSLR-2*; Wilcoxon rank-sum test) differences in the distribution of favorable and unfavorable gene expression changes in A549 cells indicate that levels of these lncRNAs may impact LUAD patient prognosis.

Evaluation of whether PSLR expression correlates with patient survival in LUAD showed that LUAD patients within the high lncRNA expression group of either of the two PSLRs had significantly (log-rank

test p<7.0 × 10$^{-3}$) increased overall survival compared with the patients that had low PSLR expression (*Figure 7B*). Moreover, high expression of the PSLR had significant favorable prognostic associations for multiple cancer types (*Figure 7B*). Our results show the cell proliferation/growth-suppressive function of the newly identified PSLRs significantly contributes to patient prognosis across multiple cancers. Altogether, our systems-level analysis provides a highly confident resource to identify previously unrecognized proliferation/growth-regulating lncRNAs that impact cancer.

## Discussion

Analysis of cell fitness after genome-wide single-gene perturbation across hundreds of cell lines enabled the identification of essential protein-coding genes, co-essential protein complexes and pathways, and proliferation/growth-regulating functions of poorly characterized genes (*Blomen et al., 2015*; *Hart et al., 2015*; *McFarland et al., 2018*; *Meyers et al., 2017*; *Tsherniak et al., 2017*; *Wainberg et al., 2021*). In contrast, only a very limited number of loss-of-function screens that covered a small number of cell lines has left many of the thousands of lncRNAs in the human genome that are essential for proliferation/growth unidentified and uninvestigated in cancer (*Cai et al., 2020*; *Liu et al., 2017a*). Like other noncoding RNAs, lncRNAs participate in complex interaction networks of genes and proteins that often have wide-spectrum effects in cell biology, including cancer cell biology (*Anastasiadou et al., 2018*). Historically, network and systems biology approaches from our group and other researchers have aided in the discovery of critical protein-coding and noncoding genes in cancer and other diseases (*Anastasiadou et al., 2018*; *Jacobsen et al., 2013*; *Kim et al., 2022*; *de Goede et al., 2021*; *Mitra et al., 2020*; *Mitra et al., 2017*; *Zhao et al., 2016*). Here, our systematic and statistically robust mapping of lncRNAs to CRISPR-derived genome-wide essential gene networks prioritizes known proliferation regulators as the top hits. Additionally, experimental validations of our prediction results demonstrated that our approach accurately assigned growth-regulating functions of previously uncharacterized lncRNAs and revealed key lncRNAs that mediate G2 cell cycle arrest. Collectively, our integrated computational and experimental framework provides a new resource for further investigations of cancer dependency on a comprehensive set of functionally uncharacterized lncRNAs.

Our network analysis includes directions in the statistically significant associations between lncRNAs and co-essential modules, further elucidating a deeper understanding of which lncRNAs are potential proliferation/growth inducers and which are suppressors. Notably, identified proliferation/growth-inducing and -suppressing lncRNAs have a strong trend to be reduced and elevated, respectively, in cells treated with anti-neoplastic drugs compared to control cells or in patient samples with lower proliferation activity compared with a higher proliferation activity. These data indicate that the observed lncRNA alterations in cancer cells may be selected for to induce or allow cell cycle progression. Our multilevel verifications from literature searches, analysis of completely independent pan-cancer validation datasets, and follow-up biological experiments indicate that the proliferation/growth-regulating lncRNAs were identified with high confidence. We obtained such high-confidence results likely due to employing our multivariate computational model that minimizes the impact of copy number changes and DNA methylation in constructing lncRNA-mRNA association networks. Secondly, we devised a statistically robust systems-level analysis aiming to capture specific co-essential modules whose alterations are associated with significant modulation of Hallmark proliferation signatures, which could directly impact cellular fitness in cancer.

In the co-essential network modules, the p53 tumor suppressor is negatively associated with many genes essential in cell cycle progression (*Feeley et al., 2017*; *Wainberg et al., 2021*). CRISPR screens show p53 wild-type cell lines gain a growth advantage upon p53 suppression (*Tsherniak et al., 2017*). When activated, the p53 tumor suppressor transcriptionally regulates genes that coordinately modulate various cellular processes that inhibit tumorigenesis by blocking cell cycle progression, protecting genome stability, and promoting apoptosis (*Levine, 2020*). Efforts over decades identified core protein-coding genes that p53 regulates across cell types and stresses and, more recently, tissue types, but these have not fully explained the effects of p53 (*Fischer, 2017*; *Levine, 2020*; *Moyer et al., 2020*; *Nguyen et al., 2018*). Recently, p53 was shown also to regulate the transcription of lncRNAs (*Chaudhary and Lal, 2017*). Here, we gained critical insights into which proliferation/growth-regulating lncRNAs are p53-regulated. Several of the proliferation/growth-suppressive lncRNAs we identified are known p53 targets (*PICART1* [*Cao et al., 2017*], *LINC00475* [*Melo et al., 2016*], *NEAT1*

[*Mello et al., 2017*], *LNCTAM34A/GUARDIN* [*Hu et al., 2018*], and *ST7-AS1* [*Sheng et al., 2021*]). However, none of the known p53-effector lncRNAs were predicted as proliferation/growth inducers, as would be expected, and indicate our statistically robust network and systems biology approach selects true positives and rejects false-positive regulations. Two top proliferation/growth-suppressor lncRNAs (*PSLR-1* and *PSLR-2*) uncovered by our analyses are two previously unstudied lncRNA whose promoters are bound by p53, as determined by multiple ChIP-seq datasets. We show that downregulation of either of these two core lncRNAs whose promoters are bound by p53 in multiple cancer types results in worse overall patient survival. Following a variety of agents that activate p53 and that are used as cancer treatments (*Levine, 2020*), we observed the induction of *PSLR-1* and *PSLR-2* in large-scale data using a robust meta-analysis. Although different sample sizes have the potential to bias meta-analysis results, there is no outlier (Tukey's fences test) in the distribution of sample sizes (*Supplementary file 2b*) across the 14 RNA-seq data that we used for the meta-analysis. Furthermore, our validation experiments on *PSLR-1* and *PSLR-2* indicate that the meta-analysis accurately predicted transcriptional changes in the cells treated by different p53-activating agents. Therefore, our combined bioinformatics and experimental results revealed two poorly characterized lncRNAs whose p53-mediated transcriptional regulation and roles in the regulation of cancer cell fitness have not been previously studied.

Our accumulated results show that p53-mediated transcriptional regulation of PSLRs correlates with modulation of multiple key genes in the p53-p21-DREAM-CDE/CHR signaling pathway (*Fischer et al., 2016*). We determined a strong enrichment of E2F4 and FOXM1 transcription factors and the *CHR* promoter motif in the PSLR-suppressed genes. These genes regulate the G2 cell cycle, and our data showed that increased levels of either of the *PSLR* lncRNAs resulted in modulation of multiple key G2-regulating genes at the RNA level and this translated to changes at the protein level and a G2 cell cycle arrest *in LUAD cells.* Moreover, there were significantly altered proliferation pathways, oncogenic and tumor-suppressor genes, and prognostic genes with alteration of PSLRs, confirming their proliferation/growth-regulatory roles. Combined, the results indicate that the ability of cells to induce these lncRNAs following stress-inducing stimuli, including DNA damaging cancer treatments, reduces their survival/growth, prolonging the survival of the patient.

Overall, the 30 pan-cancer high-confidence proliferation/growth-regulating poorly characterized lncRNAs we identified constitute an immediately useful resource for cancer biologists and other researchers. Many important future investigations will stem from this resource, including determination of transcriptional and post-transcriptional regulation of the identified lncRNAs, exploration of these lncRNAs as competitive endogenous RNA in co-essential pathways, and discovering their epigenetic roles in regulating cancer cell proliferation/growth. Altogether, our study provides a significant leap forward in understanding co-essential pathways and their constituent lncRNAs that critically modulate cellular fitness.

## Materials and methods

**Key resources table**

| Reagent type (species) or resource | Designation | Source or reference | Identifiers | Additional information |
|---|---|---|---|---|
| Cell line (*Homo sapiens*) | A549, lung adenocarcinoma | ATCC | CCL-185 | |
| Cell line (*H. sapiens*) | H460, lung adenocarcinoma | ATCC | HTB-177 | |
| Transfected construct (*Homo sapiens*) | *PSLR-1* | VectorBuilder | *FAM198B-AS1* VB200405-4575kbk | Lentiviral construct to express the lncRNA |
| Transfected construct (*H. sapiens*) | *PSLR-2* | VectorBuilder | *SEPTIN9-DT* VB200405-4635tnb | Lentiviral construct to express the lncRNA |
| Transfected construct (*H. sapiens*) | *MALAT1* | Addgene | #118580 | Lentiviral construct to express the lncRNA |
| Transfected construct | GIPZ lentiviral empty vector control | Dharmacon/Horizon | #RHS4349 | Lentiviral construct used as control |
| Antibody | Anti-cyclin B1 (mouse monoclonal) | BD Pharmingen | #554177 | WB (1:1000) |
| Antibody | Anti-14-3-3-σ (goat polyclonal) | Santa Cruz | #sc-7681 | WB (1:500) |

*Continued on next page*

*Continued*

| Reagent type (species) or resource | Designation | Source or reference | Identifiers | Additional information |
|---|---|---|---|---|
| Antibody | Anti-p21 (mouse monoclonal) | Santa Cruz | #sc-6246 | WB (1:250) |
| Antibody | Anti-cyclin A2 (rabbit polyclonal) | Cell Signaling | #91500 | WB (1:1000) |
| Antibody | Anti-GADD45A (rabbit polyclonal) | Cell Signaling | #4632 | WB (1:1000) |
| Antibody | Anti-CDK1 (mouse monoclonal) | Cell Signaling | #9116 | WB (1:1000) |
| Antibody | Anti-CDC25C (rabbit polyclonal) | Cell Signaling | #4688 | WB (1:1000) |
| Antibody | Anti-B-ACTIN (mouse monoclonal) | Sigma | #A5316 | WB (1:5000) |
| Antibody | Anti-histone H3 (phospho S10) (rabbit polyclonal) | Abcam | #ab5176 | Intracellular staining FACS (1:100) |
| Sequence-based reagent | Hsa-CDK1-For | This paper | qRT-PCR primer | GCCGGGATCTACCATACCC |
| Sequence-based reagent | Hsa-CDK1-Rev | This paper | qRT-PCR primer | AGGAACCCCTTCCTCTTCAC |
| Sequence-based reagent | Hsa-CCNB1-For | This paper | qRT-PCR primer | ATGACATGGTGCACTTTCCTCC |
| Sequence-based reagent | Hsa-CCNB1-Rev | This paper | qRT-PCR primer | GCCAGGTGCTGCATAACTGG |
| Sequence-based reagent | Hsa-CCNA2-For | This paper | qRT-PCR primer | GAAAACCATTGGTCCCTCTTG |
| Sequence-based reagent | Hsa-CCNA2-Rev | This paper | qRT-PCR primer | GGCTGTTTCTTCATGTAACCC |
| Sequence-based reagent | Hsa-CDC25C-For | This paper | qRT-PCR primer | CTTCCTTTACCGTCTGTCCAG |
| Sequence-based reagent | Hsa-CDC25C-Rev | This paper | qRT-PCR primer | CCAAGTTTCCATTGTCATCCC |
| Sequence-based reagent | Hsa-CDKN1A-For | This paper | qRT-PCR primer | GGAAGACCATGTGGACCTGT |
| Sequence-based reagent | Hsa-CDKN1A-Rev | This paper | qRT-PCR primer | GGATTAGGGCTTCCTCTTGG |
| Sequence-based reagent | Hsa-GADD45A-For | This paper | qRT-PCR primer | CCCTGATCCAGGCGTTTTG |
| Sequence-based reagent | Hsa-GADD45A-Rev | This paper | qRT-PCR primer | GATCCATGTAGCGACTTTCCC |
| Sequence-based reagent | Hsa-SFN-For | This paper | qRT-PCR primer | GAGACACAGAGTCCGGCATT |
| Sequence-based reagent | Hsa-SFN-Rev | This paper | qRT-PCR primer | CTGCCATGTCCTCATAGCGT |
| Sequence-based reagent | Hsa-BAX-For | This paper | qRT-PCR primer | CCCGAGAGGTCTTTTTCCGAG |
| Sequence-based reagent | Hsa-BAX-Rev | This paper | qRT-PCR primer | CCAGCCCATGATGGTTCTGAT |
| Sequence-based reagent | Hsa-AchR-For | This paper | qRT-PCR primer | AGGACCCTACAGACCCCTCTTC |
| Sequence-based reagent | Hsa-AchR-Rev | This paper | qRT-PCR primer | AGTGTTCATGGTGGCTAGGTG |
| Sequence-based reagent | Hsa-PSLR1-For | This paper | qRT-PCR primer | AGCCACCCTTGACTGAGGTA |
| Sequence-based reagent | Hsa-PSLR1-Rev | This paper | qRT-PCR primer | CTCTGTCTTCTGTGCCGTGT |

*Continued*

| Reagent type (species) or resource | Designation | Source or reference | Identifiers | Additional information |
|---|---|---|---|---|
| Sequence-based reagent | Hsa-PSLR2-For | This paper | qRT-PCR primer | GAAAGTCAGCGATTCCTGCG |
| Sequence-based reagent | Hsa-PSLR2-Rev | This paper | qRT-PCR primer | GTTGGACAGTTCCTCCCTGAG |
| Sequence-based reagent | Hsa-B-ACTIN-For | This paper | qRT-PCR primer | CACCAACTGGGACGACAT |
| Sequence-based reagent | Hsa-B-ACTIN-Rev | This paper | qRT-PCR primer | ACAGCCTGGATAGCAACG |
| Commercial assay or kit | RT2 SYBR Green ROX qPCR Mastermix | QIAGEN | #330521 | qPCR Mastermix |
| Commercial assay or kit | SuperScript III First-Strand Synthesis System | Invitrogen | #18080051 | cDNA synthesis |
| Commercial assay or kit | TRIzol Reagent | Invitrogen | #15596026 | RNA isolation |
| Chemical compound, drug | MTT (3-(4,5-Dimethyl-2-thiazolyl)–2,5-diphenyl-2H-tetrazolium bromide) | Sigma-Aldrich | #475989 | Used for proliferation assay |
| Chemical compound, drug | Crystal violet | Sigma-Aldrich | #192 | Used for colony formation assay |
| Chemical compound, drug | Propidium iodide | Sigma-Aldrich | #537059 | Used for cell cycle analysis |
| Chemical compound, drug | Nutlin-3 | Sigma-Aldrich | #444143 | p53-activating stimuli |
| Chemical compound, drug | Etoposide | Sigma-Aldrich | #341205 | p53-activating stimuli |
| Chemical compound, drug | Cisplatin | Sigma-Aldrich | #232120 | p53-activating stimuli |
| Chemical compound, drug | Colcemid | Gibco | #15212012 | Used to treat cells for positive control for mitotic cells |
| Software, algorithm | FlowJo | TreeStar | | Dean–Jett–Fox cell cycle model used for cell cycle analysis |
| Software, algorithm | GraphPad Prism | GraphPad Prism (https://graphpad.com) | PRID:SCR_002798 | |
| Software, algorithm | R Project for Statistical Computing | R Project for Statistical Computing (https://www.r-project.org/) | PRID:SCR_001905 | |

## CCLE and TCGA multi-omics data analysis

We extracted gene copy number, DNA methylation, and mRNA expression profiles of 807 cancer cell lines from CCLE (*Ghandi et al., 2019*) and 3017 patient samples of 10 TCGA cancer types from the UCSC Xena database (*Goldman et al., 2020*). In brief, gene-level copy number variation was measured using the GISTIC2 or ABSOLUTE method (*Mermel et al., 2011*). For DNA methylation, data from reduced representation bisulfite sequencing (for CCLE) or Illumina Infinium HumanMethylation450 platform (for TCGA) were used, except for TCGA ovarian cancer, for which we used Illumina Infinium HumanMethylation, as a larger set of DNA methylation profiles was available. In TCGA, relative DNA methylation levels, described as β-values, ranged from 0 to 1. The values represent the ratio of the intensity of the methylated bead type to the combined locus intensity. If more than one methylation probe mapped to a gene promoter region, a representative probe was selected that showed the strongest negative Spearman's rank correlation between methylation β-value and mRNA expression of that gene (*Jacobsen et al., 2013*; *Mitra et al., 2017*). In the mRNA-seq expression profiles, mRNAs were selected for downstream analysis if their normalized expression values were ≥1 in at least 75% of the samples. Additionally, we downloaded processed lncRNA expression profiles for the same 10 TCGA cancer types from the Tanric database (*Li et al., 2015*). For CCLE, we extracted the annotated lncRNA expression profiles from the mRNA-seq data. To select lncRNAs for downstream analysis, we

employed two filtering criteria as described previously (*Mitra et al., 2017*; *Yan et al., 2015*). First, we eliminated lncRNAs if the 50th percentile of normalized expression values was not >0; and second, we selected lncRNAs if the 90th percentile of the normalized expression values was >0.1.

## Multivariate regression to predict lncRNA-mRNA associations

For each gene, we first calculated z-scores for its promoter methylation, mRNA/lncRNA expression, and copy number profiles (*Ghandi et al., 2019*). We estimated whether lncRNA ($x_{\text{lnc}_l}$) and mRNA ($Y$) expression association in $n$ cell lines (for CCLE data) or $n$ tumor samples of a given cancer type (for TCGA data) was independent of DNA methylation ($x_{\text{DM}}$) and copy number variation ($x_{\text{CNV}}$) events using the following multivariate regression analysis:

$$Y_s = \beta_0 + \beta_{\text{DM}}x_{\text{DM},s} + \beta_{\text{CNV}}x_{\text{CNV},s} + \beta_{\text{lnc}_l}x_{\text{lnc}_l,s} \quad , \; s = 1, \, \ldots, \, n$$

Here, $\beta_0$ is the intercept. Regression coefficients $\beta_{\text{DM}}$, $\beta_{\text{CNV}}$, and $\beta_{\text{lnc}_l}$ indicate the association strength between RNA-level expression change of the given gene with DNA methylation, copy number, and lncRNA expression changes, respectively. The BH adjusted regression p<0.001 was used as the cutoff to select significantly associated lncRNA-mRNA pairs in CCLE or a specific TCGA cancer type.

## Mapping lncRNAs to co-essential modules

Genome-wide co-essential modules (n = 5218) were obtained from *Wainberg et al., 2021*. The modules were generated using three different module density cutoffs 0.2, 0.5, and 0.9. The module density score determines how strong the internal connections are within a module. We retained 2223 non-syntenic modules (i.e., all genes in the module are not on the same chromosome) with a density score of 0.5 or 0.9. The Gene Ontology database was used by Wainberg et al. to assign functions of individual modules, which may not reflect comprehensive and diverse oncogenic functions that the module members may coordinately regulate. We, therefore, reannotated module functions using cancer Hallmark genesets and the KEGG pathways, available in Molecular Signature Database (*Subramanian et al., 2005*). The module functions were annotated with top five Hallmark signatures and top five KEGG pathways among the significantly (BH adjusted hypergeometric test p<0.05) enriched signature/pathway list. The enrichment analysis was performed using the R package WebGestaltR (*Zhang et al., 2005*). We finally selected 123 modules that show significant enrichment of proliferation/growth-regulating genesets/pathways.

We merged multivariate regression model-derived lncRNA-mRNA associations with proliferation/growth-regulating essential modules based on the genes commonly present in the two networks. We assigned an lncRNA to a module if the module members were significantly (BH adjusted hypergeometric test p<0.05) enriched with the lncRNA-associated protein-coding genes and the lncRNA-module association strength was significantly stronger (BH adjusted 1000 randomization test p<0.05). The average regression coefficient scores, measured from the lncRNA and module members, were used to predict the lncRNA-mediated co-essential module regulation direction. We predicted lncRNA-mediated positive (induction) or negative (suppression) regulation of co-essential modules based on the calculated positive and negative average regression scores. The final network was constructed from 1027 lncRNA, 78 proliferation/growth-linked co-essential modules, and 34,226 edges that connect one lncRNA to a module.

## Meta-analysis of lncRNA and co-essential module associations

For each proliferation/growth-linked co-essential module, we grouped the lncRNAs based on their positive and negative associations with the module, respectively. We ranked each group of lncRNAs based on their average regression scores with the module members, which indicates lncRNA-module association strength. For 78 proliferation/growth-linked modules, we obtained rankings of 78 sets of positively and 78 sets of negatively associated lncRNAs. Obtained rankings were subjected to a probabilistic, nonparametric, and rank-based method RobustRankAggreg V1.1 (*Kolde et al., 2012*) to identify the lncRNAs with consistently positive or negative associations with the modules. The method assigns p-values as a significance score to indicate whether one achieves consistently better ranking under the null hypothesis that the ranked lncRNA lists are uncorrelated. We repeated the same analysis 78 times after excluding one of the rankings. Acquired p-values were averaged to assess

the stability of the significance scores and further corrected by BH method (*Benjamini and Hochberg, 1995*) to minimize potential false positives. We considered the lncRNAs to be high-confidence proliferation/growth regulators if they had consistent positive- or negative association with statistical significance (BH corrected robust rank aggregation p<0.05) (*Benjamini and Hochberg, 1995*).

## Analysis of RNA-sequencing data

We evaluated 14 RNA-seq datasets from Gene Expression Omnibus (GEO) (*Barrett et al., 2013*) database (*Supplementary file 2b*). Each dataset consists of samples treated by p53-activating growth inhibitors and DMSO/untreated controls. Seven datasets contain samples treated with Nutlin-3, and the remaining were treated with doxorubicin, 5-fluorouracil, or ionizing radiation in one of the 10 different cell lines representing eight different cell types. To check the quality of the sequencing reads, fastq files of RNA-seq data were analyzed with FASTQC (version 0.11.8; https://www.bioinformatics.babraham.ac.uk/projects/fastqc/). The script infer_experiment.py from R package RSeQC was used to determine strand specificity (*Wang et al., 2012*), and adapter sequences were trimmed using Cutadapt version 2.4 (*Martin, 2011*). We aligned reads to the human genome (Ensembl version GRCh38) using Hisat2 (*Kim et al., 2019*) and assembled transcripts with StringTie version 1.3.6 (*Pertea et al., 2016*). A Python script prepDE.py from StringTie was used to extract read count information. The read counts were imported into edgeR (*Robinson et al., 2010*), a negative binomial distribution was used to normalize the data, and differential expression of lncRNA and protein-coding genes was determined in the p53-activated cells compared with control cells (*Robinson et al., 2010*).

## Meta-analysis of lncRNA and gene expression

Differentially expressed lncRNAs and protein-coding genes determined from individual RNA-seq datasets were grouped into up- and downregulated and ranked based on fold-change expression values. From 14 publicly available RNA-seq datasets, treated with p53-activating growth inhibitors, we obtained rankings of 14 sets of up- and 14 sets of downregulated genes. To identify consistently dysregulated lncRNAs and genes, obtained rankings were subjected to RobustRankAggreg V1.1 (*Kolde et al., 2012*). We repeated the same analysis 14 times after excluding one of the rankings. Acquired p-values were averaged to assess the stability of the significance scores and further corrected by BH method (*Benjamini and Hochberg, 1995*) to minimize potential false positives.

## Determining favorable and unfavorable genes/lncRNAs

To determine overall survival-associated genes or lncRNAs in a TCGA cancer type, we ranked the patients according to the expression of a specific gene/lncRNA. We divided the sample into low or high groups based on the median expression cutoff value of the gene/lncRNA. Using univariate survival analysis with a log-rank statistical test, we evaluated which genes/lncRNAs in that specific TCGA patient cohort have a significant (|Z-score| > 1.96 that corresponds to p<0.05; log-rank test) impact on patient survival (*Gentles et al., 2015*). The genes/lncRNAs were divided into favorable or unfavorable prognostic groups depending on whether their overexpression significantly increases or reduces patient overall survival.

## Determining proliferation activity across TCGA patient samples

For each TCGA patient sample, we averaged the expression values of the genes present in the KEGG cell cycle and Hallmark proliferation/growth-regulating genesets (MYC targets V1 and V2, E2F targets, G2/M checkpoint, and mitotic spindle). For each cancer type, we ranked the samples based on the calculated average expression values, denoted as proliferation activity scores. The lowest and highest quartile samples were designated as the lower and higher proliferation activity groups, respectively.

## Cell culture, vectors, and viral production

LUAD cell lines (H460 and A549 from ATCC) were cultured in RPMI-1640 media supplemented with 10% FBS, 100 U/mL penicillin, and 100 μg/mL streptomycin. A549 media was also supplemented with additional L-glutamine. Cell lines were authenticated by STR profiling by ATCC and confirmed to be free of mycoplasma. Lentiviral vectors encoding *FAM198B-AS1* (*PSLR-1*) or *SEPTIN9-DT* (*PSLR-2*) were purchased from Vector Builder. The lentiviral vector encoding human *MALAT1* was purchased from Addgene (plasmid #118580). All vectors were sequenced to verify the lncRNA encoded. Lentivirus

was produced by transient transfection into 293T cells using the calcium phosphate precipitation method. One million LUAD cells were placed into 10 cm plates 16 hr prior to infection. Also, 16 hr after infection, cells were harvested and plated for further experimental analyses (see below).

## RNA isolation, qRT-PCR, and RNA-sequencing

A549 and H460 cells ($1 \times 10^5$ cells/well) in 6-well plates were treated with 10 μM etoposide, cisplatin, or Nutlin-3 (all three from Sigma) or received 5 Gy of gamma radiation (cesium source), and were harvested at intervals. A549 and H460 cells expressing lncRNA *PSLR-1* and *-2* or vector only via lentiviral infection were harvested 48 hr after infection (see above). Total RNA was isolated using TRIzol (Ambion) according to the manufacturer's protocol, except samples were incubated in isopropanol overnight at –20°C to increase RNA enrichment. cDNA was generated using the SuperScript III First-Strand Synthesis System (Invitrogen), expression of lncRNA and mRNA was determined, in triplicate, using SYBR Green qPCR Mastermix (QIAGEN), and values normalized to *β-ACTIN* and presented as $2^{-\Delta Ct}$, $2^{-\Delta\Delta Ct}$, or log2 fold change (indicated in the figure legend). Primer sequences are given in ***Supplementary file 2d***. For RNA-sequencing, quadruplicate samples of A549 cells expressing lncRNA *PSLR-1* or *-2* or vector only were harvested 48 hr after lentiviral infection. RNA-seq libraries were prepared using NEBNext Ultra II RNA library preparation kit. Raw RNA-seq profiles generated from Illumina HiSeq 4000 were obtained from GENEWIZ (https://www.genewiz.com/en) and data analyzed as described above.

## Cell proliferation

A549 and H460 cells infected with lentiviral vectors expressing lncRNA *PSLR-1, -2* or *MALAT1* or empty vector were placed in 96-well plates (2500 cells/well), in quadruplicate. As soon as cells attached (within 8 hr), the 0 hr reading was measured by MTT proliferation assay (562 nm; Sigma) according to the manufacturer's protocol. MTT assays were then performed every 24 hr thereafter for 4 days.

## Colony formation assay

A549 and H460 cells infected with lentiviral vectors expressing lncRNA *PSLR-1, -2* or *MALAT1* or empty vector were cultured at low density (200 cells/well), in triplicate, in 6-well plates. After 14 days in culture, colonies were stained with 0.5% crystal violet and counted (≥50 cells defined a colony) using an inverted microscope. Pictures were taken using the camera feature of an LG-G6 phone with 13MP standard-angle lens.

## Cell cycle and viability analyses

A549 and H460 cells infected with lentiviral vectors expressing lncRNA *PSLR-1, -2* or *MALAT1* or empty vector were cultured ($1 \times 10^5$ cells/well) in 6-well plates (one sample per condition per time point). Cells were harvested at intervals and stained with propidium iodide to evaluate DNA content by flow cytometry as described (***Adams and Eischen, 2014***). The Dean–Jett–Fox model on FlowJo (TreeStar) was used to quantify the percentage of cells in each phase of the cell cycle. Experiments were repeated four independent times for both cell lines. Intracellular staining for phosphorylated histone H3 (phospho-S10; Abcam) was repeated twice and was evaluated using the manufacturer's protocol to detect cells in mitosis; cells treated with colcemid (0.05 μg/mL for 16 hr; Gibco) served as a positive control for cells in mitosis. Live cell number and viability were determined using Trypan Blue dye exclusion 48 hr after placing cells in 6-well plates. One sample of each condition from four independent experiments was quantified for both cell lines.

## Western blotting

A549 and H460 cells expressing lncRNA *PSLR-1* and *-2* or vector only were harvested 48 hr after lentiviral infection, and whole-cell protein lysates were obtained as previously published (***Eischen et al., 1999***). Antibodies include CCNA2 (E1D9T, #91500), GADD45A (D17E8, #4632), CDK1 (POH1, #9116), and CDC25C (5H9, #4688) from Cell Signaling Technology; CDKN1A (F-5, sc-6246) and 14-3-3-σ (N-14, sc-7681) from Santa Cruz Biotechnology; CCNB1 (GNS-1, #554177) from BD Pharmingen, and β-ACTIN (AC-74, #A5316) from Sigma.

## Patient samples

Deidentified non-small-cell LUAD patient samples and normal lung tissue were obtained from the Vanderbilt University Medical Center Lung Biorepository. Samples were acquired from surgical resections and were collected with patient consent. A board-certified pathologist verified by H&E stained sections that the samples were >80% tumor for tumor samples or lacked evidence of precancerous lesions for normal tissue samples.

## Statistics

Unpaired $t$-tests were conducted to determine differentially expressed lncRNAs and protein-coding genes except for the RNA-seq data that contain read counts. For that, we performed edgeR (*Robinson et al., 2010*). All statistical tests were two-tailed, except for computing combined p-values using Fisher's method because it requires the input data to be derived from one-tailed test. The statistical tests used in individual analyses are indicated in the text and/or figure/table legends. All statistical tests with p-values of <0.05 were considered statistically significant. Experiments with more stringent p-value cutoffs were explicitly indicated. For the biological experiments (MTT, colony, cell number, cell cycle, and qRT-PCR), unpaired two-tailed $t$-tests were used when comparing two groups.

## Acknowledgements

We thank members of the Eischen lab for helpful discussions. This work was supported by the National Cancer Institute R01CA177786 (CME), National Cancer Institute Cancer Center core grant P30CA056036 that supports the MetaOmics and flow cytometry core facilities of the Sidney Kimmel Cancer Center, the Steinfort gift (CME), and the Herbert A Rosenthal endowed chair (CME).

## Additional information

### Competing interests

Christine M Eischen: receives research funding from AbbVie unrelated to this manuscript. The other authors declare that no competing interests exist.

### Funding

| Funder | Grant reference number | Author |
|---|---|---|
| National Cancer Institute | R01CA177786 | Christine M Eischen |
| National Cancer Institute | P30CA056036 | Christine M Eischen |

The funders had no role in study design, data collection and interpretation, or the decision to submit the work for publication.

### Author contributions

Ramkrishna Mitra, Conceptualization, Data curation, Formal analysis, Investigation, Methodology, Validation, Visualization, Writing – original draft; Clare M Adams, Investigation, Methodology, Writing – original draft; Christine M Eischen, Conceptualization, Formal analysis, Funding acquisition, Methodology, Supervision, Writing – original draft

### Author ORCIDs

Ramkrishna Mitra http://orcid.org/0000-0002-6181-9066
Clare M Adams http://orcid.org/0000-0002-9940-3795
Christine M Eischen http://orcid.org/0000-0003-4618-8996

### Decision letter and Author response

Decision letter https://doi.org/10.7554/eLife.77357.sa1
Author response https://doi.org/10.7554/eLife.77357.sa2

## Additional files

### Supplementary files

• Supplementary file 1. (a) Long noncoding RNAs (lncRNAs) that have significantly enriched positive associations with essential genes. (b) lncRNAs that have significantly enriched negative associations with essential genes. (c) Reannotation of co-essential modules using cancer Hallmark gene set enrichment analysis. Genes in the individual modules can be obtained from PubMed ID: 33859415. (d) Reannotation of co-essential modules using Kyoto Encyclopedia of Genes and Genome (KEGG) pathway enrichment analysis. Genes in the individual modules can be obtained from PubMed ID: 33859415. (e) Number of co-essential modules significantly enriched with indicated KEGG pathways or Hallmark genesets. Rankings of individual pathways/genesets were based on number of co-essential modules. (f) Networks of lncRNA and co-essential modules. Co-essential module details can be obtained from PubMed ID: 33859415. (g) Meta-analysis-derived potential proliferation/growth-inducing lncRNAs. (h)Meta-analysis-derived potential proliferation/growth-suppressing lncRNAs. (i) Number of pubmed hits for individual lncRNAs. PubMed was searched on September 11, 2021. (j) lncRNAs that have more than twofold expression change in one or more cancer types. (k) Proliferation/growth-suppressing lncRNAs showing consistent upregulation across RNA-seq data. (l) Proliferation/growth-inducing lncRNAs showing consistent downregulation across RNA-seq data.

• Supplementary file 2. (a) Differential distribution of expression changes of proliferation inducing and suppressing long noncoding RNAs (lncRNAs) across TCGA cancer types. (b) Summary of the 14 RNA-seq datasets evaluated. (c) Summary of the 17 ChIP-seq datasets evaluated. (d) Primers used for qRT-PCR.

• Transparent reporting form

### Data availability

All data generated or analyzed during this study are included in the manuscript and supporting files, except RNA-sequencing data, which has been deposited into GEO - https://www.ncbi.nlm.nih.gov/geo/query/acc.cgi?acc=GSE169186. We used 43 publicly available data sets (see Supplementary files 2b and 2c cited in text).

The following dataset was generated:

| Author(s) | Year | Dataset title | Dataset URL | Database and Identifier |
|---|---|---|---|---|
| Eischen CM, Mitra R, Adams CM | 2022 | Analysis of gene expression in lung adenocarcinoma cells following ectopic expression of proliferation-suppressive lncRNAs (PSLR) | https://www.ncbi.nlm.nih.gov/geo/query/acc.cgi?acc=GSE169186 | NCBI Gene Expression Omnibus, GSE169186 |

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
