## [Editor Report]

This manuscript makes a valuable contribution to the common understanding of the function of lncRNAs in cancer formation and progression. Besides developing and applying a robust analysis framework of large-scale pan-cancer omics datasets to discover the roles of 30 long non-coding RNAs (lncRNAs) in cancer proliferation and growth, the authors performed direct function-testing experiments to validate the predicted biological mechanisms of two lncRNAs. The analysis framework developed here can serve as a resource to study the functions of lncRNA in cancer, and the computational framework can also be further extended to study cancer-relevant transcriptional and post-transcriptional regulation.

---

## [Decision Letter]

**Decision letter after peer review:**

Thank you for submitting your article "Systematic lncRNA mapping to genome-wide co-essential pathways uncovers cancer dependency on uncharacterized lncRNAs" for consideration by *eLife*. Your article has been reviewed by 2 peer reviewers, and the evaluation has been overseen by a Reviewing Editor and Erica Golemis as the Senior Editor. The following individuals involved in the review of your submission have agreed to reveal their identity: Xi Steven Chen (Reviewer #1); Xingyi Guo (Reviewer #2).

Essential revisions:

1. The modeling work described on Page 5 lines 108-114, and Page 6 is a bit confusing. Were those two analyses the same or independent? On page 5, 540 positives and 500 negative lncRNAs were discovered, but 1027 lncRNAs were identified on Page 6. It's important to clarify the analysis workflow here. It's also necessary to mention the total number of annotated lncRNAs used in the analysis.

2. On page 8, the top 200 negatively and 200 positively associated genes for each lncRNA were used for downstream analysis, which was an arbitrary cutoff. Instead, the Kolmogorov-Smirnov test implemented in GSEA could be considered to use the whole gene list to test enrichment.

3. Could the authors clarify the reason 50 Hallmark gene signatures were used in TCGA analysis instead of using the same co-essential modules in CCLE analysis?

4. Could the authors clarify the number of TCGA samples used in the analysis? For example, we know there are more than 1000 BRCA samples, but only 526 samples were used in the analysis. In 7B, why were only a few cancer types listed instead of all ten cancer types?

5. In the multivariate regression, the authors should include a covariant, 'cancer type' if they used all data combined from different cancer types, to adjust potential cancer heterogeneity. Alternatively, they can perform such analysis for each cancer.

6. In lines 530-532: "The BH adjusted regression P<0.001was used as the cut-off to select significantly associated lncRNA-mRNA pairs in CCLE or a specific TCGA cancer type." To identify significantly correlated lncRNA-mRNA pairs, the authors need to justify why they would not just use results from TCGA to replicate the associations.

7. In lines 547-549: "We finally selected the modules that show significant enrichment of proliferation/growth-regulating genesets/pathways." -The authors need to provide a detailed number here.

8. In lines 555-557: "0.05. The average regression coefficient scores, measured from the lncRNA and module members, were used to predict the lncRNA-mediated co-essential module regulation direction." The authors need to discuss the limitation of this strategy to predict the regulation direction, as the statistical approach is not well justified.

9. In the Methods section, "Meta-analysis of lncRNA and gene expression", the authors need to discuss the limitation of this approach, as the different sample sizes could lead to a significant bias. The authors should consider that the results could be more reliable from the largest sample size of RNA-seq dataset.

10. The authors may wish to include additional analyses using data for lung squamous cell carcinomas, as they are available in TCGA.

*Reviewer #2 (Recommendations for the authors):*

I hope my comments below could help authors improve their work in their revision.

Well-established cancer-driver genes/cancer gene consensus may need to be considered for their annotations.

---

## [Author Response]

Essential revisions:1. The modeling work described on Page 5 lines 108-114, and Page 6 is a bit confusing. Were those two analyses the same or independent?

The two analyses on page 5 and page 6 were not independent and apologize that this was confusing. The 1049 lncRNAs identified from the analysis on page 5 were used to evaluate their association with co-essential modules in the analysis on page 6. In the revised manuscript, on page 6, we have clarified this.

On page 5, 540 positives and 500 negative lncRNAs were discovered, but 1027 lncRNAs were identified on Page 6. It's important to clarify the analysis workflow here. It's also necessary to mention the total number of annotated lncRNAs used in the analysis.

Here is the detailed analysis workflow with the number of lncRNAs filtered in each step. We started with 15777 lncRNAs, annotated in GENCODE (V27) database. Of them, 1049 (549 and 500 lncRNAs with positive and negative associations, respectively) have significantly enriched associations with the pancancer cell survival/growth-regulating essential genes that we identified from the analysis on page 5. Among the 1049 lncRNAs, 1027 lncRNAs showed significant associations with proliferation/growthregulating co-essential modules that we identified from the analysis on page 6. To increase clarity on the number of lncRNA at each of these steps, we revised the manuscript on pages 5 and 6.

2. On page 8, the top 200 negatively and 200 positively associated genes for each lncRNA were used for downstream analysis, which was an arbitrary cutoff. Instead, the Kolmogorov-Smirnov test implemented in GSEA could be considered to use the whole gene list to test enrichment.

As suggested, considering the whole gene list, we implemented GSEA to evaluate lncRNA and Hallmark gene signature associations across the ten TCGA cancer types. We compared the new results with the results shown in Figure 2A in the manuscript, where we considered the top 200 negatively and top 200 positively associated genes. Both the results indicate that our predicted proliferation/growth-linked lncRNAs preferentially regulate the Hallmark category proliferation across TCGA cancer types compared to the other Hallmark categories. Furthermore, the distribution of the lncRNAs across eight

broad categories of Hallmark processes also consistently shows significant (*P*<8.69x10^-14^; ANOVA) variability in the number of regulating lncRNAs. Consistent biological interpretation from the results of two different approaches and our follow-up experimental validation results indicate that the results obtained from the top 200 genes reflect the true biological associations between the lncRNA and cancer Hallmark processes. We revised the manuscript with this additional analysis on page 9 and added the data to the supplement (Figure 2—figure supplement 1). The data further strengthen our prediction results, and we thank Reviewer 1 for raising this question.

3. Could the authors clarify the reason 50 Hallmark gene signatures were used in TCGA analysis instead of using the same co-essential modules in CCLE analysis?

Here we aimed to identify lncRNAs that have a consistent association with proliferation/growthregulating functions both in cancer cell lines and patient samples. Our objective was to measure the consistency/reproducibility using completely independent training and test data sets. Hence, we utilized CCLE data and co-essential modules as the training data and TCGA and cancer Hallmark data as the independent test data. On page 21, we stated “Our multi-level verifications from literature searches, analysis of completely independent pan-cancer validation datasets, and follow-up biological experiments indicate that the proliferation/growth-regulating lncRNAs were identified with high confidence.” Moreover, the 50 Hallmark gene signatures include various cancer gene sets with different oncogenic functions. These high-quality gene signatures enabled us to additionally investigate if the cell line-derived predicted proliferation/growth-regulating lncRNAs preferentially regulate proliferation in the cancer patient samples compared to other oncogenic mechanisms (see Figure 2A and Figure 2—figure supplement 1).

4. Could the authors clarify the number of TCGA samples used in the analysis? For example, we know there are more than 1000 BRCA samples, but only 526 samples were used in the analysis.

The multivariate regression model included DNA copy-number, DNA methylation, mRNA, and lncRNA expression data. Only the samples from which all these four types of omics data were generated were selected for the analysis. This resulted in the reduction of sample size for each cancer type. To improve clarity, we have added Venn diagrams for all ten cancer types in Figure 1—figure supplement 2 indicating how we obtained the number of samples in Figure 1—figure supplement 1 that we used for the multivariate regression analysis. We have also revised the figure legend of Figure 1—figure supplement 1.

In 7B, why were only a few cancer types listed instead of all ten cancer types?

Only the cancer types showing a significant correlation between the indicated lncRNA expression and overall patient survival were reported in Figure 7B.

5. In the multivariate regression, the authors should include a covariant, 'cancer type' if they used all data combined from different cancer types, to adjust potential cancer heterogeneity. Alternatively, they can perform such analysis for each cancer.

We apologize that it was not clear in the manuscript that we performed the multivariate regression analysis for each of the ten TCGA cancer types separately. To clarify this in the revised manuscript, we edited the following sentence on page 9,

“To assess the impact of the cell line-derived proliferation/growth-linked lncRNAs in cancer patient samples; we evaluated lncRNA-mRNA associations independently in each of the 10 TCGA cancer types utilizing the same multivariate regression model (Methods section; Figure 1A).”

6. In lines 530-532: "The BH adjusted regression P<0.001 was used as the cut-off to select significantly associated lncRNA-mRNA pairs in CCLE or a specific TCGA cancer type." To identify significantly correlated lncRNA-mRNA pairs, the authors need to justify why they would not just use results from TCGA to replicate the associations.

There may be a misunderstanding on this point so we will clarify it here. Our objective in this study was to identify and verify reproducible proliferation/growth-regulating lncRNAs in different cellular contexts and not to evaluate the specific lncRNA-target gene pairs. To identify the proliferation/growth-regulating lncRNAs, we evaluated different data sets from cell lines, tumor tissues, and across cancer types. Literature mining verification and follow-up validation experiments showed that our approach accurately identified proliferation/growth-regulating lncRNAs. Identifying replicated lncRNA-gene associations would be useful in a future study to specifically investigate high-confidence lncRNA-target genes.

7. In lines 547-549: "We finally selected the modules that show significant enrichment of proliferation/growth-regulating genesets/pathways." -The authors need to provide a detailed number here.

In the revised manuscript, we added to this sentence that 123 modules were selected (page 29).

8. In lines 555-557: "0.05. The average regression coefficient scores, measured from the lncRNA and module members, were used to predict the lncRNA-mediated co-essential module regulation direction." The authors need to discuss the limitation of this strategy to predict the regulation direction, as the statistical approach is not well justified.

Our computational framework introduced regulation direction to predict with confidence whether one lncRNA induces or suppresses cancer cell proliferation/growth. With a meta-analysis and statistical tests, we gained such confidence. For the reviewer’s convenience, we provide the following details that are stated in the Material and Methods in sub-section “Meta-analysis of lncRNA and co-essential module associations”, on page 30.

“Meta-analysis of lncRNA and co-essential module associations

For each proliferation/growth-linked co-essential module, we grouped the lncRNAs based on their positive and negative associations with the module, respectively. We ranked each group of lncRNAs based on their average regression scores with the module members, which indicates lncRNA-module association strength. For 78 proliferation/growth linked modules, we obtained rankings of 78 sets of positively and 78 sets of negatively associated lncRNAs. Obtained rankings were subjected to a probabilistic, non-parametric, and rank-based method RobustRankAggreg V1.1 (Kolde et al., 2012) to identify the lncRNAs with consistently positive or negative associations with the modules. The method assigns *P*-values as a significance score to indicate whether one achieves consistently better ranking under the null hypothesis that the ranked lncRNA lists are uncorrelated. We repeated the same analysis 78 times after excluding one of the rankings. Acquired *P*-values were averaged to assess the stability of the significance scores and further corrected by Benjamini-Hochberg (BH) method (Benjamini & Hochberg, 1995) to minimize potential false positives. We considered the lncRNAs to be high-confidence proliferation/growth regulators if they had consistent positive- or negative association with statistical significance (BH corrected robust rank aggregation *P*<0.05) (Benjamini & Hochberg, 1995).”

9. In the Methods section, "Meta-analysis of lncRNA and gene expression", the authors need to discuss the limitation of this approach, as the different sample sizes could lead to a significant bias. The authors should consider that the results could be more reliable from the largest sample size of RNA-seq dataset.

This reviewer is correct that different sample sizes could lead to a bias. Therefore, after conducting a Tukey’s fences test, in the revised manuscript, we added to page 22, “Although different sample sizes have the potential to bias meta-analysis results, there is no outlier (Tukey’s fences test) in the distribution of sample sizes (Supplementary Table 11) across the 14 RNA-seq data that we used for the meta-analysis. Furthermore, our validation experiments on *PSLR-1* and *PSLR-2* indicate that the meta-analysis accurately predicted transcriptional changes in the cells treated by different p53 activating agents.”

10. The authors may wish to include additional analyses using data for lung squamous cell carcinomas, as they are available in TCGA.

We specifically chose 10 different cancer types originating from different organs for our study. We did not include lung squamous cell carcinoma (LUSC) because we had chosen to include the most frequent type of lung cancer, lung adenocarcinoma (LUAD). However, in response to this Reviewer’s suggestion, we evaluated TCGA LUSC data and compared the results with LUAD data. We determined the number of predicted proliferation/growth-regulating lncRNAs that have significantly (FDR < 0.05; Gene Set Enrichment Analysis) enriched association with different cancer hallmark categories in LUAD and LUSC (see Author response image 1). We observed a strong correlation (Spearman’s r = 0.84) between LUAD and LUSC in the number of lncRNAs that potentially regulate the indicated Hallmark categories. Specifically, proliferation is associated with the highest number of lncRNAs in both subtypes of lung cancer. Therefore, the LUSC data are similar to that of LUAD data, which further supports our conclusion that the identified lncRNAs have proliferation regulation potential across different cancer types.

**Author response image 1. sa2fig1:** Number of predicted proliferation/growth-regulating lncRNAs that have significant (FDR < 0.05; GSEA) enrichment with the indicated cancer hallmark categories in LUAD and/or LUSC. P-value was derived from Spearman’s rank correlation test.

Reviewer #2 (Recommendations for the authors):I hope my comments below could help authors improve their work in their revision.

Addressing this Reviewer's comments did improve the overall quality and clarity of our work. We addressed all the Reviewer’s comments point-by-point below.

Well-established cancer-driver genes/cancer gene consensus may need to be considered for their annotations.

As recommended, we included the oncogenes and tumor-suppressor genes available in the cancer gene census of COSMIC database to our analysis. Previously, in Figure 4C, we evaluated fold-change distribution of literature curated, high-confidence oncogenes and tumor-suppressor genes in the A549 lung adenocarcinoma cells with elevated levels of either *PSLR-1* or *PSLR-2* compared to vector control treated cells. In the revised version, we added the oncogenes and tumor-suppressor genes present in the cancer gene census to the existing literature curated gene set. We updated Figure 4C and edited page 14, in the Results section and page 36 in the data availability section in the revised manuscript to include this additional analysis.